

# Technical note: an alternative water vapor sampling technique for stable isotope analysis

César Dionisio Jiménez–Rodríguez[1,2], Miriam Coenders–Gerrits[1], Thom Bogaard[1], Erika Vatiero[1,3], and Hubert Savenije[1]

[1]Delft University of Technology. Water Resources Section. Stevinweg 1, 2628 CN Delft, The Netherlands.
[2]Tecnológico de Costa Rica. Escuela de Ingeniería Forestal. 159-7050, Cartago, Costa Rica.
[3]Università degli Studi della Campania Luigi Vanvitelli. Department of Environmental, Biological and Pharmaceutical Sciences and Technologies. Via Vivaldi, 43-81100. Caserta, Italy.

**Correspondence:** César Dionisio Jiménez–Rodríguez (cdjimenezcr@gmail.com)

**Abstract.** Recent developments in laser spectroscopy enabled to carry out direct measurements of $\delta^2$H and $\delta^{18}$O of air water vapor in the field. However, certain experimental sites or project budgets do not ease the deployment of this technology to obtain the needed measurements. We carried out three consecutive experiments aiming to provide an alternative method to sample air vapour in the field, and preventing fractionation during the process. The first experiment determined the minimum air sample volume required to obtain measurements of $\delta^2$H and $\delta^{18}$O with a laser spectrometer. The second one test evaluated the capacity to retrieve continuously similar isotopic signatures of the collected samples from one location. The third experiment assessed the applicability of this methodology under an experimental set up in a coniferous forest in The Netherlands. Stable isotope measurements of water vapor by laser spectroscopy can be obtained with a sample volume of 450 mL of air. This allows to measure each sample during a period of 300 s, obtaining isotope signatures with standard deviations lower than 0.1‰ and 0.5‰ for $\delta^{18}$O and $\delta^2$H, respectively. Air samples collected with bags were homogeneously mixed, allowing to retrieve a better temporal variation in the field than the cold traps employed.

## 1 Introduction

Evaporation after precipitation is the largest flux in the hydrological cycle (Coenders-Gerrits et al., 2014; Miralles et al., 2011; Wallace, 1995; Wang et al., 2014) and its precise estimation is therefore critical for improving hydrological models and water management decisions (Guo et al., 2017). This flux is composed of water vapor from plant transpiration, and evaporation from soil and water intercepted by plant and litter surfaces; being their partitioning a key element to understand the evaporation process at different time and spatial scales (Blyth and Harding, 2011; Lawrence et al., 2007; Raghunath, 2006; Savenije, 2004; Wang and Yakir, 2000; Dubbert et al., 2017).

The introduction of stable water isotopes as conservative tracers allows for a better understanding of the individual components that form the evaporation at local and global scale (Barbour, 2017; Coenders-Gerrits et al., 2014; Jasechko et al., 2013; Kool et al., 2014). Both stable isotopes $\delta^2$H and $\delta^{18}$O are considered ideal tracers in hydrology because of their natural occur-





rence and signature variation linked to the isotope ratio dependency on temperature (West et al., 2006). Water phase change drives physical isotope fractionation, as well as diffusion and mixing but at lower proportions (Gat et al., 2000; Rothfuss et al., 2010). Evaporation from soil and wet surface depends on the amount of water vapor transferred towards the atmosphere, and will undergo physical fractionation (Soulsby et al., 2017; Sprenger et al., 2017; Van Bavel and Hillel, 1976). Whilst the frac-

tionation caused during photosynthesis is an example of the chemical fractionation, determining the isotope signature of the water transpired by plants (Butt et al., 2010; Cernusak et al., 2016; Farquhar et al., 2007). Physical and chemical fractionation of water stable isotopes characterizes the kinetic fractionation process, which is unidirectional and mass dependent (Young et al., 2002).

Isotope signatures of water vapor that originated from evaporation are often estimated with the Craig-Gordon model (CG-model) (Craig and Gordon, 1965). This model was first developed to determine the water vapor signature of evaporation coming from open waters (Horita et al., 2008). The model distinguishes three interface layers between the water surface and the atmosphere. The transition layer is located at the liquid boundary layer, were equilibrium fractionation is the dominant process. It is followed by the diffusion layer, where the molecule transport is governed by the molecular diffusivity. The last

layer is characterized by non–fractionation and turbulent mixing with the air layer above the water (Craig and Gordon, 1965; Gat, 2008; Horita et al., 2008; Roden and Ehleringer, 1999). The CG-model includes the equilibrium effect from the phase change from liquid to vapor and the kinetic effect due to the diffusion of the heavy and light isotopes of water vapor in air (Roden and Ehleringer, 1999).

Several studies have applied different modifications to the CG-model, using it to estimate soil evaporation and transpiration, however, some assumptions may not strictly be valid in both cases considering the sensitivity of $\delta^{18}O$ with temperature (Dubbert et al., 2013). Transpired water vapor depends on the plant water status as a consequence of wetting–drying episodes, environmental conditions such as diurnal cycles and water uptake (Barbour, 2007; Ferrio et al., 2009; Williams et al., 2004). Therefore, differences up to tens of ‰ can be found in both isotope signatures (Horita et al., 2008) driving discrepancies

between the model and the real leaf signatures as well as soil water signatures (Roden and Ehleringer, 1999; Barbour, 2007).

Another method to assess isotope signatures of water vapor is through isotope measurements of sampled air vapor collected with cold traps. These ones condensate and freeze the water vapor in a canister immersed in liquid nitrogen or dry ice, thawing it after collection to retrieve a liquid sample (He and Smith; Kool et al., 2014; Sheppard, 1958; Wen et al., 2016). This process

has a high risk of fractionation due to phase changes, leading to heavier isotope signatures if the collection efficiency is not perfect (Griffis, 2013). This type of bias can be found even in phase changes in close systems as the cryogenic extractions, where is not possible to recover the full signature of a known sample (Orlowski et al., 2018).

Recent developments in measurement techniques for water stable isotopes showed the capacity of mass spectrometers and

off-axis integrated cavity output spectroscopy (OA-ICOS) techniques to measure air water vapor directly from sealed contain-



ers with moist soil during short periods of time to determine the soil water isotope signature (Hendry et al., 2015; Wassenaar et al., 2008). Now with improvements in OA-ICOS technology direct measurements of air vapor on the field can be done, skipping the use of cold traps during the experimental setups (Rambo et al., 2011; Steen-Larsen et al., 2013, 2014; Steen-Larsen et al., 2015). This system provides reliable measurements of stable isotope signatures of water vapor when the setup includes a

water vapor isotope standard source (WVISS) device and the application of corrections for the concentration dependency and temporal drift (Kurita et al., 2012). However, when it is not possible to deploy this type of equipment in the field (e.g., no field cabin or enclosure, no controlled run temperatures, no constant power supply) or budget restrictions the need of new sampling techniques to collect air vapor with no fractionation arise.

This work aims to provide a field vapor sampling method where the samples can be analyzed in the laboratory using laser spectroscopy with an OA-ICOS unit, preventing fractionation during its collection and analysis. We provide a procedure to determine the minimum sample volume required to obtain a reliable isotope signature from vapor samples and we compare this with cold trap sampling procedure.

## 2   Methodology

### 2.1   Instrumentation

A Water Vapor Isotope Analyzer (WVIA; model 912) was used to determine the isotope signature of water vapor samples, with the support of a LGR Water Vapor Isotope Standard Source (WVISS; model 908-0004-9002) employed to provide a controllable flow of water vapor with a known liquid standard measurement for an absolute calibration of raw measurements. The device pumping rate for all the samples is fixed at $\sim 90 \, \mathrm{mL \, min^{-1}}$. The WVIA and the WVISS were attached to a Multiport

Inlet Unit (MIU; model: LGR 908-0003-9002) for the automatic control of eight inlets to measure multiple samples for specific periods of time. The MIU has eight ports for 6 mm diameter tubing which allows the development and attachment of different sampling devices. In all the measurements, the first MIU inlet was attached to a dried air source to facilitate the data analysis providing a different air signature with a simultaneous dryer condition. This dried air source was achieved by filling a 2 L borosilicate bottle with 1.5 kg of silica gel to dry the laboratory air to a concentration lower than 5000 ppm. All measurements

were performed at a sampling interval of 5 s, obtaining the average and standard deviation of each measurement.

### 2.2   Experimental design

In order to determine the reliability of measuring stable isotopes of water vapor from small air samples, we established three consecutive experiments to estimate the accuracy of the method and the possible sources of error; building every experiment on the results of the previous one. The first experiment (Section 2.2.3) aimed to determine the minimum air volume required

to obtain measurements of $\delta^2 H$ and $\delta^{18} O$ with standard deviations lower than 1.5 ‰ and 0.30 ‰ respectively to ensure reliable measurements (Kurita et al., 2012). The second experiment (Section 2.2.4) focused on the capacity to retrieve similar isotope



signature of samples collected from one location continuously, and to identify differences among different locations. Finally, the third experiment (Section 2.2.5) assessed the applicability of this methodology in an experimental set up measuring differences in $\delta^2$H and $\delta^{18}$O signatures along the vertical profile of a coniferous forest in The Netherlands. All analysis and statistical test were performed with the software R (R Core Team, 2017).

## 2.2.1 Measurements

Isotope signatures ($^2$H and $^{18}$O) of both sample types (liquid and gas) expressed in respect to the Vienna Standard Mean Ocean Water (VSMOW) following equation 1 (Craig, 1961). Liquid samples were measured through the injection of $900\,\mu$L into a heating chamber for complete vaporization of the water and flushed into the IWA. The measurement procedure for gas samples will be described later in the description per experiment (Sections 2.2.3 – 2.2.5).

$$\delta = \left(\frac{R_{\text{sample}}}{R_{\text{standard}}} - 1\right) \times 1000\%o \tag{1}$$

Where $\delta$ is the relative concentration (‰) of the stable isotope $^2$H or $^{18}$O, $R$ is the stable isotope ratio ($^2$H/$^2$1 or $^{18}$O/$^{16}$O) of the standard water ($R_{\text{standard}}$) and the sample ($R_{\text{sample}}$).

## 2.2.2 Water Vapor Calibration

Isotope signature measurements of water vapor depends on the concentration of water molecules (ppm) and the specific drift of the OA-ICOS unit, which makes it essential to calibrate each individual measurement (Aemisegger et al., 2012; Rambo et al., 2011; Kurita et al., 2012; Steen-Larsen et al., 2013, 2014). The calibration of water vapor measurements was performed automatically with the WVISS. This procedure was based on the direct measurement of water vapor with a known isotopic signature ($\delta^2$H: -40.8±0.86 ‰ and $\delta^{18}$O: -5.60±0.09 ‰) using the WVISS unit. The calibration was performed with three selected water molecule concentrations depending on the air sample concentrations (Sections 2.2.3 – 2.2.5), ensuring to cover the total range of water molecules concentration along the sample measurements (green dots in Figure 1). A calibration run is performed before every seven samples after which the seven bags with air samples were manually replaced on the MIU and a new sequence of measurements started. The raw signatures of $\delta^2$H and $\delta^{18}$O are calibrated using the correction factors ($\alpha_{\text{O}}$ and $\alpha_{\text{H}}$) determined based on the dependency of raw signatures to the water mixing ratio ($w$). The polynomial coefficients $a$, $b$ and $c$ in the equations 2 and 3 were determined for every set of measurements per experiment (Rambo et al., 2011; Kurita et al., 2012; Steen-Larsen et al., 2013, 2014). The calibrated value of each stable isotope ($\delta^{18}$O and $\delta^2$H) is determined with equations 4 and





5 where the $\delta^{18}O_{raw}$ and $\delta^2H_{raw}$ are the raw measurements given by the device (Rambo et al., 2011; Steen-Larsen et al., 2013).

$$\alpha_O = \frac{\delta^{18}O_{raw}}{\delta^{18}O_{known}} = a_O w^2 + b_O w + c_O \qquad (2)$$

$$\alpha_H = \frac{\delta^2H_{raw}}{\delta^2H_{known}} = a_H w^2 + b_H w + c_H \qquad (3)$$

$\quad \delta^{18}O = \frac{1}{\alpha_O}\delta^{18}O_{raw} \qquad (4)$

$$\delta^2H = \frac{1}{\alpha_H}\delta^2H_{raw} \qquad (5)$$

### 2.2.3 Experiment 1: Response Time

A set of seven time intervals (60 s, 120 s, 180 s, 240 s, 300 s, 450 s, and 600 s) were selected to determine the minimum time required to obtain a stable measurement with a standard deviation of 1.5‰ and 0.30‰ for $\delta^2H$ and $\delta^{18}O$, respectively. This

minimum time will determine the minimum sample volume required to get a stable measurement of ambient air sampled from an open inlet of the MIU. In order to identify the memory effect from a previous analysis, ambient air from the laboratory and dried air were measured alternately with the same time intervals (Fig.1). The experiment run six times, for a total sampling period of almost 8 hr. The calibration of raw measurements during this experiment was performed with a water vapor concentration of 4600 ppm, 6500 ppm and 8350 ppm (Appendix A1).

A moving average window of one minute was applied to all the data sets to determine the minimum time required to obtain a reliable signature with standard deviations lower than 1.5‰ in $\delta^2H$ and 0.30‰ in $\delta^{18}O$. Equation 6 was used to determine the Allan variance ($\sigma_A^2$) of the calibrated signatures of $\delta^{18}O$ and $\delta^2H$. This analysis determines the time interval from where the moving average is driven by the laser white noise and not by the memory effect of the previous sample. This analysis evaluates

the difference between consecutive measurements ($y_i$ and $y_{i+1}$) aggregated at the same time interval ($\tau$) and averaged over the total number of measurements ($n$) (Aemisegger et al., 2012; Allan, 1966). The square root of the Allan variance ($\sigma_A^2$) corresponds to the Allan deviation ($\sigma_A$), that will be used as parameter to define the data variation driven by the device white noise (Aemisegger et al., 2012). When we plot $\sigma_A$ against the time interval ($\tau$), is possible to determine when the measurement's variation is driven by white noise of the device and not by changes due to different air sample signatures (Aemisegger et al.,

2012). This point is characterized by a plateau or a constant value in the plot.





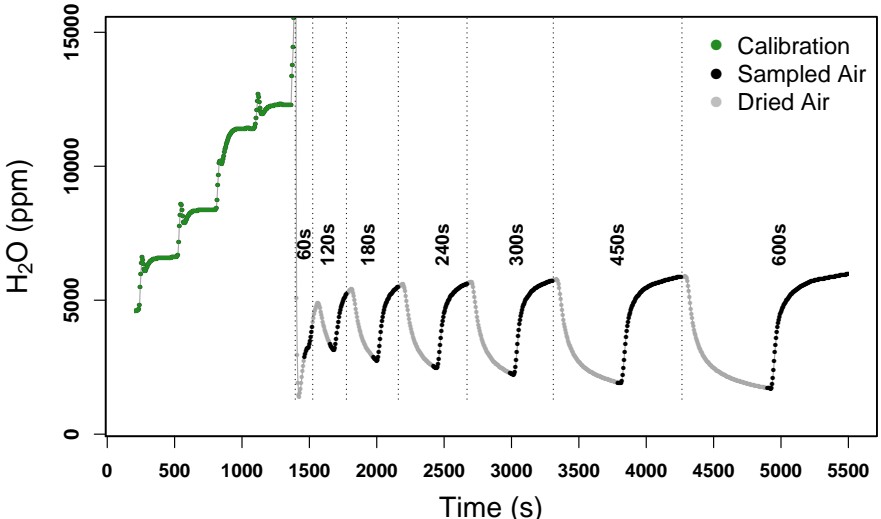

**Figure 1.** Water vapor concentration (ppm) during the calibration (green points) and raw measurements of sampled (black dots) and dried air (grey dots). Every set of measurements was performed at 60 s, 120 s, 180 s, 240 s, 300 s, 450 s, and 600 s for both types of air.

$$\sigma_{\mathrm{A}}^2(\tau) = \frac{1}{2n} \sum_{i=1}^{n} \left( y_{i+1}(\tau) - y_i(\tau) \right)^2 \tag{6}$$

### 2.2.4 Experiment 2: Consistency

From experiment 1 we learned that an air sample of 450 mL (equal to 300 s measuring) can provide stable measurements of isotope signatures with standard deviations lower than 1.5 ‰ and 0.30 ‰ for $\delta^2$H and $\delta^{18}$O, respectively. Therefore, we se-
lected a commercially available polyethylene bag of 1.1 L used for filling packaging spaces as a sampling container for every air sample. The extra volume allowed us to redo the measurements if it was required. The sampling bags are fabricated with a simple valve made from polyethylene as well. Thus, the valve is tightly closed with the inner air bag pressure reducing the risk of sample contamination during its transport and storage (Fig. 2). A special inlet was built to connect each sampling bag to the individual ports of the MIU. The inlet is composed of four parts allowing to plug in and out each sample bag without mixing
the air contained in the bag with the laboratory air (Fig. 2). Parts A, B and D convey the air flow from the sample bag towards the MIU. Part A connects the inlet to the different MIU ports, it has an outer diameter of 6 mm. Part B has a outer diameter of 4 mm and allows the tight movement of the part C along all its length. Part C has a conical shape of 5 mm to 9 mm of outer diameters. This part seals the valve opening preventing to lose air from the sampling bag and allow to adjust to the length of the sampling bag valve. Part D has a conical shape of 1 mm to 5 mm of outer diameters, allowing to inset the full extension of
the inlet within the sampling bag valve.





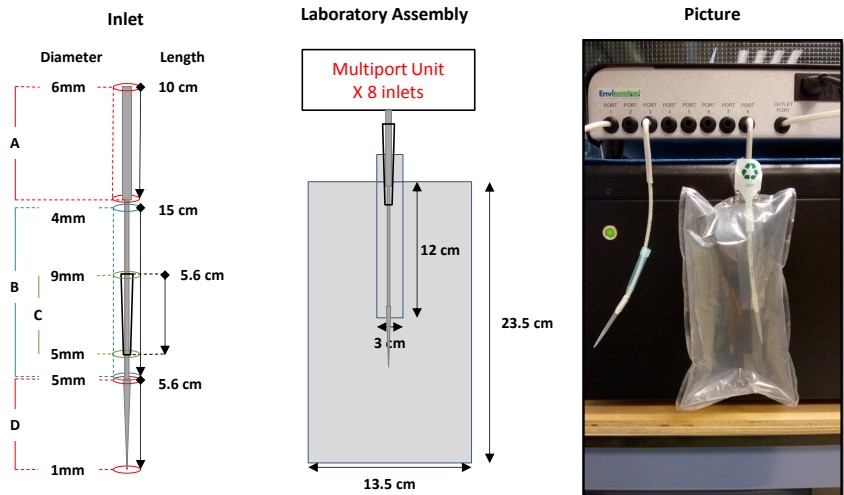

**Figure 2.** Laboratory assembly details of the Multiport Inlet Unit (MIU) connection, sampling bag and the view of the inlet connection with the multiport unit.

Manual air samples were collected with a medical cardiopulmonary resuscitation (CPR) balloon with a conical plastic inlet that allows to push the air into the sample bag. Four sets of samples were collected at different locations and ambient conditions within The Netherlands. The air samples were collected at Speulderbos (Sample A), Delft outdoors (Sample B and C), laboratory air (Sample D). During the measurements, we sampled the laboratory air directly to determine if the air from the laboratory affects the sample measurements. Each sample was measured during 5 min in the OA-ICOS device and the calibration of raw measurements was performed with the following water vapor concentrations: 6500 ppm, 8350 ppm and 11400 ppm (Appendix A2). Deuterium excess ($D_e$) was determined with equation 7, based on the relationship between $\delta^2H$ and $\delta^{18}O$ along the Global Meteoric Water Line (Rozanski et al., 1993). An analysis of variance (ANOVA) was applied to determine statistical differences (p=0.05), while the HSD-Tukey test (Tukey, 1949) allowed to differentiate among samples with different signatures.

$$D_e = \delta^2H - 8.2 * \delta^{18}O \tag{7}$$

### 2.2.5 Experiment 3: Field Validation

An experimental site located in a coniferous forest in The Netherlands (Speulderbos) was selected to carry out experiment 3. Here, we tested the capacity to identify isotope signature changes of $\delta^2H$ and $\delta^{18}O$ along the vertical axis of the forest. At the site, a flux tower of 48 m is located in the center of a 2.5 ha plot of Douglas fir (*Pseudotsuga menziesii*) stand at De Veluwe region (Cisneros Vaca et al., 2018; Schilperoort et al., 2018). One micro-meteorological station was placed in an open area of 0.68 ha at 400 m from the experimental site. This station measured at 2 m height the air temperature (°C) and relative humidity



(%) with a 12-bit Temperature/Relative Humidity, model: S-THB-M008. Incoming short wave radiation ($W\,m^{-2}$) with a silicon pyranometer (model: S-LIB-M003) and precipitation ($mm\,d^{-1}$) with a Davis rain gauge (model: S-RGD-M002).

Air vapor samples were collected at three heights along the flux tower (Fig. 3), with sampling points located below the canopy (17 m), above the canopy layer (34 m) and at the top of the tower (47 m). Air moves through a 3D printed radiation shield of 6 cm diameter and 7 cm height (Ham, 2015) adapted to support a small fan at the bottom to allow the air movement and a new top to hold a 6 mm diameter sampling tube. This device was placed on an pole extending 2.5 m East from the tower. The sampling tube at each height conveys the air along 50 m towards an air pump at the bottom of the tower, sucking the air at a rate of $3\,L\,min^{-1}$ for a travel time of less than 2 min from the sampling point to the collection point. Before sampling starts the pumps were running for 15 min to ensure a continuous mixed air flow along the entire tube length. A set of precipitation samples collected since 2016 on a monthly basis were used to determine the local meteoric water line (LMWL). The samples were gathered in a raingauge device designed following the recommendations from Gröning et al. (2012). A 15 cm diameter funnel collected the precipitation into a 5 L bottle, conveying the water towards the bottom of the bottle through a 15 cm tube of 9 mm diameter. A 5 m tubing of 6 mm diameter connect the air within the bottle and the environment to reduce the vapour exchange.

Five sets of samples were collected during four hours on 10 July 2018 from 12:00 to 16:00 h. Each set was composed of a liquid sample collected with a cold trap during one hour, while six air bag samples were filled simultaneously every 10 minutes for 1 hr. This makes a total of 18 air samples. The cold trap consists of a test tube of 30 mL immerse in dry ice (-70 °C) within an isolated cooler of 2 L volume. The air flow was pumped through a 6 mm diameter tubing of nitrile rubber (FESTO PUN-6x1). The low temperature allows the condensation and freezing of the water vapor present on the air flow. The frozen water vapor was tightly closed after the sampling period and let it thaw for 5 min, collecting immediately the liquid water sample into a 1.5 mL borosilicate glass vials for further analysis. Collected air sample bags were transported to the laboratory at TU Delft and kept at room temperature (21 °C). We checked that no water vapor condensation occurred in the sampling bag before its measurement. The analysis of these samples was performed using the following water vapor concentrations: 6500 ppm, 8350 ppm and 11400 ppm (Appendix A3).

## 3 Results and Discussion

### 3.1 Experiment 1: Response Time

To determine the minimum air volume required to obtain a stable measurement of $\delta^2H$ and $\delta^{18}O$, we selected seven time intervals to analyze laboratory air. The one minute moving averages of the calibrated signatures of $\delta^2H$ and $\delta^{18}O$ are shown in figure 4. It shows the signature variation of each time interval, where the first measurements correspond to the isotope signature of the remaining dried air in the tubing between the MIU switch valve and the measurement chamber. The steep change in both





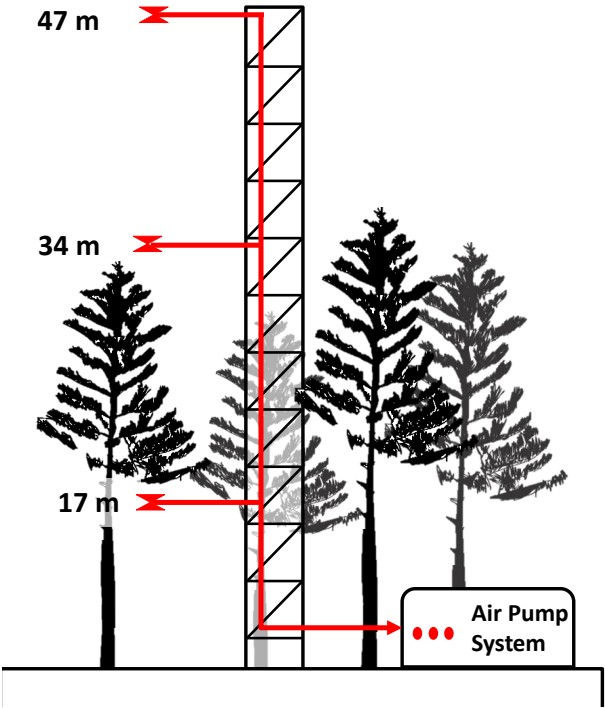

**Figure 3.** Field sampling setup established along the flux tower in Speulderbos, The Netherlands. Air sampling points are located at 2.5 m away from the tower structure to reduce the tower influence on the air flow. The air pumping system and collection points are located at the based of the tower in a non acclimatized cabin.

isotope signatures before the first 180 s corresponds to the mixing effect of the previous sample with the ambient air sampled, reaching stable isotope signatures after the 180 s of measurements. The analysis of Allan deviation values (Fig. 5), shows the strong influence of the memory effect before the first 100 s of measurements. After this measuring time, the signature variation within the same sample is linked to the white noise of the device (Aemisegger et al., 2012).

The stable isotope signatures of the sampled ambient air show differences among the five different runs as a consequence of the open system being evaluated with the MIU open inlet. The conditions within the controlled laboratory room experienced small transient variations, because of the air conditioning within the room and the residual air surrounding the device. Additionally, unexpected interruptions within the sampled room allowed air from nearby rooms to mix with the sampled air

10  leading to these differences. Between 180 s and 240 s the isotope signatures begin to stabilize with standard deviations lower than 0.3 ‰ for $\delta^{18}$O and 1.5 ‰ for $\delta^2$H), however some averages with non stable measurements in both isotopes can be found before 240 s (Figure 4). Consequently, aiming to guarantee at least one minute of stable measurements we select a time interval of 300 s to measure individual air samples. Considering the pumping rate of 90 mL min$^{-1}$ of our device, this equals to an air





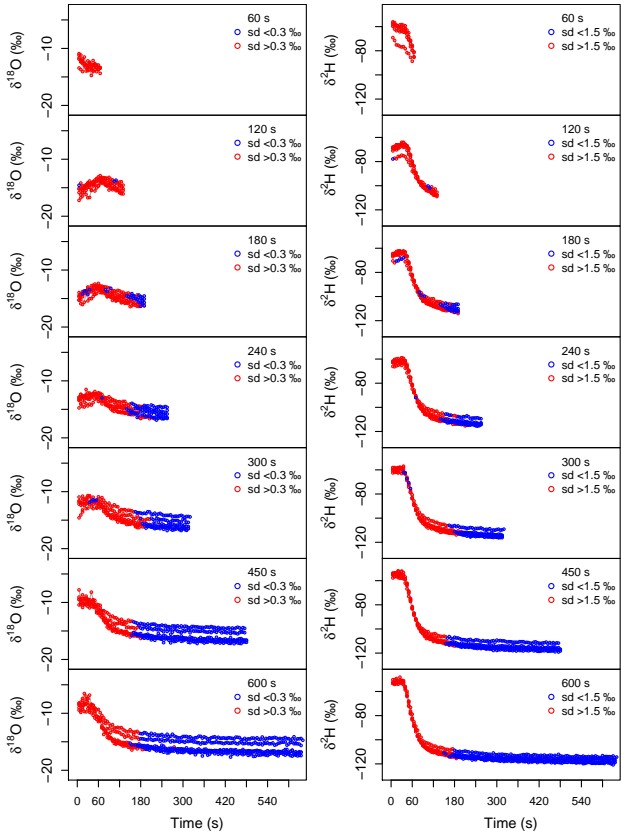

**Figure 4.** One minute moving average of the calibrated isotope signatures during the five sets of measurements during the experiment 1. The standard deviation threshold of 0.3‰ and 1.5‰ for $\delta^{18}O$ and $\delta^2H$, respectively defines the moving averages with standard deviations above the threshold (red dots) as non stable and below the threshold (blue dots) as stable measurements.

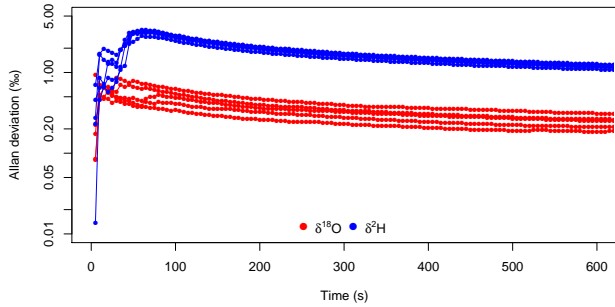

**Figure 5.** Allan deviation ($\sigma_A$) for the mobile averages of $\delta^{18}O$ and $\delta^2H$ during the 10 minutes sampling time of experiment 1.

sample of 450 mL to carry out 300 s of continuous measurements.





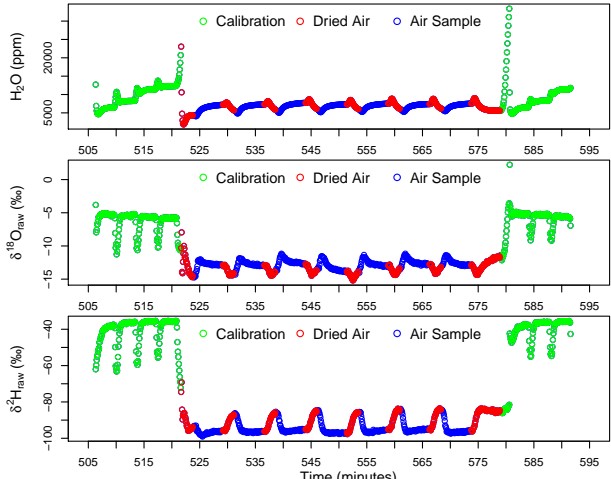

**Figure 6.** Example of uncalibrated sample results for seven air samples and the correspondent calibration test at different $H_2O$ molecule concentrations.

## 3.2 Experiment 2: Consistency

This experiment verifies whether we can retrieve the same isotope signature from a parcel of air with this technique. Air samples collected with the 1.1 L bags were able to provide a stable signature after two minutes of analysis (Fig. 6). This time is shorter than the obtained 180 s from experiment 1 (Section 3.1), because the close bags are perfectly mixed while the room air experienced small transient variations. The variation provided by the source of dried air in between samples allow to identify among individual samples and to observe the capacity to reduce the memory effect from the previous air sample.

As the isotopic signature of the laboratory air deviated from the air samples collected in plastic bags (Table 1), we have a good indication that there is no mixing between air in the laboratory and in the samples bags when flushed into the IWA. This is supported by the statistical differences showed by the $\delta^{18}O$, $\delta^2H$ signatures and $D_e$. Samples B and C were collected the same day which is the reason why $D_e$ do not show differences between them, while Sample A differed in $D_e$ from the other three samples and the laboratory because it was collected at 150 km from Delft in a different day. Despite the high accuracy on the individual measurements and the statistical differences showed among samples, the measured signature per sample depict some scatter of the averaged isotope signature measured. For both isotope signatures the standard deviation almost doubled the recommended standard deviations of 0.3‰ and 1.5‰ (Kurita et al., 2012). This scatter can be related to wind presence, modifying slightly the isotope signature during the sampling collection which requires at least 10 min per set of samples.





**Table 1.** Stable isotope signatures and deuterium excess for all air samples collected in the experiment 2.

| Sample | $\delta^{18}$O (‰) | $\delta^2$H (‰) | $D_e$ (‰) |
|---|---|---|---|
| Laboratory (n=13) | -12.9±0.2 [a] | -103.0±0.8 [a] | 3.23±1.3[bc] |
| Sample A (n=9) | -13.6±0.5 [b] | -108.9±1.8 [cd] | 2.97±2.7[c] |
| Sample B (n=9) | -13.9±0.3 [c] | -109.3±2.3 [d] | 4.84±1.4[a] |
| Sample C (n=9) | -13.8±0.4 [bc] | -107.4±1.8 [b] | 5.45±2.2[a] |
| Sample D (n=9) | -13.7±0.3 [bc] | -107.9±1.7 [bc] | 4.72±1.8[ab] |

Note: Different lower case letters on the same column are statistically different with the Tukey test (p = 0.05).

## 3.3 Experiment 3: Field Validation

This experiment assesses the applicability of our methodology in an experimental setup in Speulderbos, The Netherlands. During the sampling on 10 July 2018, 2.8 mm of precipitation occurred between midnight and noon, with no rain events during the sampling period or after (Fig. 7). The weather conditions were dominated by the presence of clouds and a high relative humidity before the sampling start. During the first three hours of sampling the relative humidity was above 80.0 %, decreasing towards 66.3 % at the end of the sampling at 16:00 hours. This decrease in air humidity was triggered by peaks of radiation with the subsequent increment in air temperature.

Based on 34 precipitation samples collected between June 2016 and July 2018, we define a local meteoric water line (LMWL) for the experimental site at Speulderbos (Fig. 8). This line follows the relationship $\delta^2$H $= 7.350 \times \delta^{18}$O $+ 4.636$ (R$^2$: 0.9393, p<0.001), whilst the precipitation preceding the air sampling period is located in the heavier section of the LMWL ($\delta^{18}$O: -2.36‰, $\delta^2$H: -14.2‰). Water vapor from both sampling procedures, cold traps and sampling bags, are located in the lighter section of the LMWL. Cold trap samples are scattered on the left side of the $^2$H-$^{18}$O plot, with a range of 10‰ in $\delta^2$H and 5‰ in $\delta^{18}$O. Air samples collected with the sampling bags are on the right side of the LMWL, and less variation in isotope signatures among the samples. Considering the humid climate of the experimental site, the isotope signature of atmospheric air should be located on the right hand side of the LMWL (McGuire and McDonnell, 2007). This is not accomplished by the cold trap samples, which are located on the left side characterizing a dry climate not according to the experimental site. Sampling bag signatures depict an isotope signature slightly evaporated. This mixture of vapor could be originated from water evaporated from intercepted surfaces and transpired water from different sources than previous rain events.

Air vapor signatures from both types of samples differ with a higher proportion for $\delta^2$H than $\delta^{18}$O. Cold trap samples have a heavier $\delta^2$H as a consequence of incomplete condensation within the cold trap. This issue leads to the enrichment of the collected water sample and it is present during air vapor sampling with cold traps with a constant air flow (Schoch-Fischer





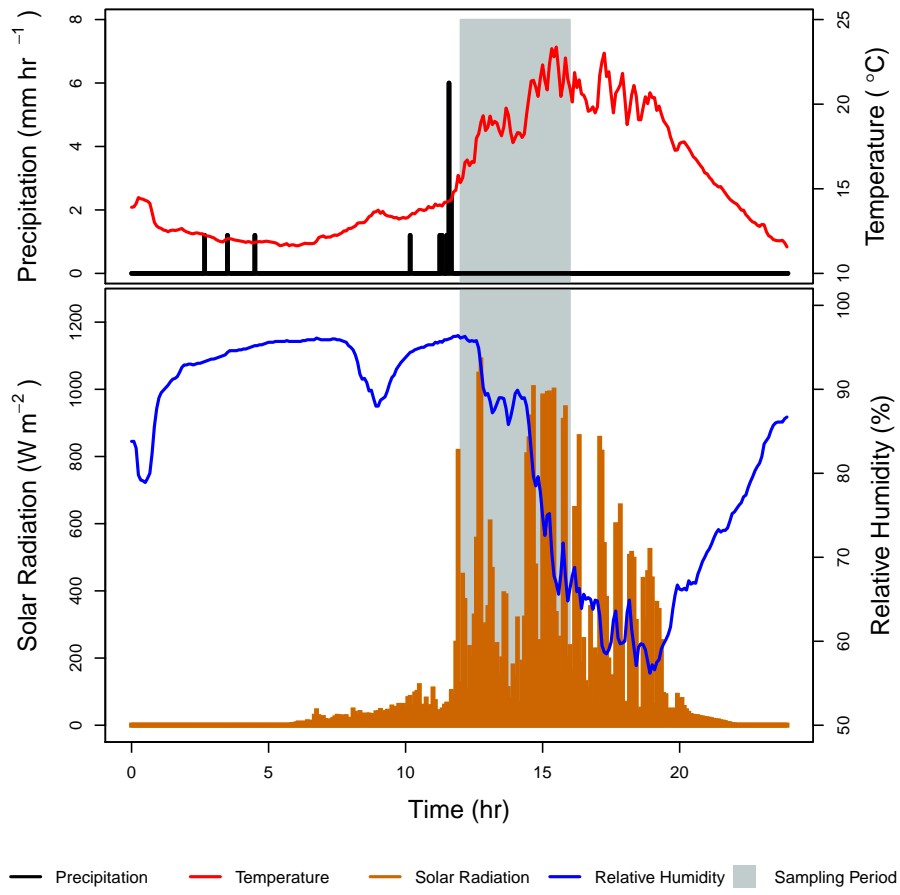

**Figure 7.** Meteorological conditions along July 10, 2018. Shaded area corresponds to the sampling period.

et al., 1984; Rhee et al., 2004) or even in closed systems during cryogenic distillation of soil samples that required the full recovery of water signatures (Orlowski et al., 2018). These differences in $\delta^2$H between cold trap and vapor samples accounts up to 30‰ during all sampling period at 47 m height and after 14:00 at 17 m (Fig. 9). The only exception is shown in the samples collected between 13:00–14:00 at 17 m height. Isotope signature at 34 m height from cold traps depicts a random pattern due to the incomplete condensation. Smaller differences in $\delta^{18}$O can be found between both types of samples, with some of the cold trap samples matching the vapor samples for small periods of time. These can be seen at 47 m before 12:00 and one gas sample at 15:00; whilst at 34 m it is seen before 12:00 and after 13:00. Cold trap samples are enriched in respect to the gas samples in no more than 4‰. Only two gas samples agreed with the cold trap signatures of $\delta^{18}$O at heigths 47 m (11:45 hours) and 34 m (13:00 hours).

Water vapor samples collected with sampling bags allow to observe the short time variability in isotope signatures due to the quick sampling time compared to the cold traps samples. The cold traps allowed to obtain a sample of almost 1 mL every





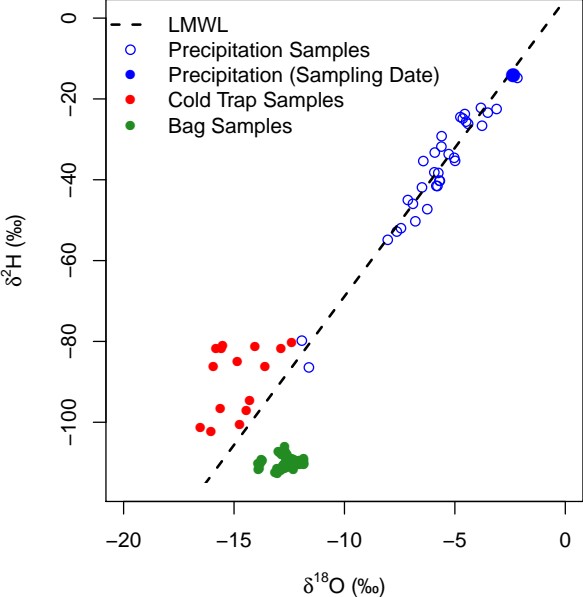

**Figure 8.** Dual isotope plot of air samples collected with sampling bags and cold traps on July 10, 2018. The local meteoric water line (LMWL) is based in 34 rain samples collected between between June 2016 and July 2018.

hour, while the gas sampling allows to increase the number of samples depending on the sampling design.

## 4 Conclusions

Experiment 1 showed that stable measurements of water vapor by laser spectroscopy can be obtained with a sample volume
of 450 mL from a air flow. This allows to measure each sample during a period of 300 s, obtaining isotope signatures with standard deviations lower than 0.1 ‰ and 0.5 ‰ for $\delta^{18}O$ and $\delta^2H$, respectively. However, with the use of sampling bags during experiment 2 was possible to get stable signatures after 180 s of measurements. Thus, because of the homogeneous mixing within the sampling bags allowing to isolate the sample from the surrounding air, preventing contamination and/or mixing during transport and analysis. The homogeneity reached by the air sample within the sampling bag allowed to retrieve
a better temporal variation than cold traps during experiment 3. Managing to get more samples per unit of time with a lower investment in supplies and better reliability than cold traps. Considering the post-processing time required per sample, the field sampling method is recommended for short sampling periods with temporal resolutions of 5 min or higher. For exploratory surveys to obtain the isotope signatures of atmospheric water before and after experiments, or as a complementary data source. For environments with high relative humidity, it is necessary to estimate the dew temperature to prevent condensation within
the sampling bags if the room temperature during storage of measurement is lower than the field.





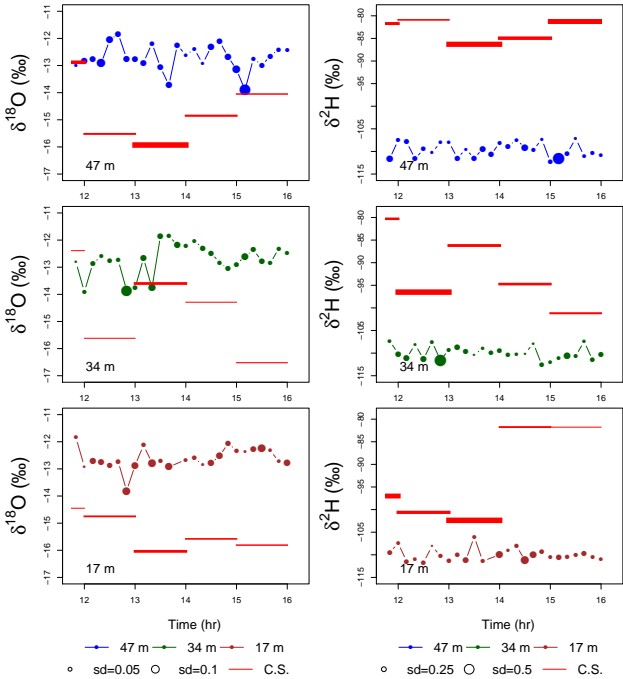

**Figure 9.** Isotope signatures obtained from gas and cold trap samples collected during experiment 3 at Speulderbos. Horizontal red lines correspond to the isotopic signature of the cold trap with the width proportional to the standard deviation of each sample.

*Acknowledgements.* This work was carried out with the aid of a grant (863.15.022) from The Netherlands Organization for Scientific Re-
search (NWO) and PINN-MICITT Costa Rica (contract: PED-032-2015-1). Special thanks to H.C. Steen-Larsen for his clarifications about
the calibration procedures. To Adriana González-Angarita (MSc) for her support during the field measurements. To Bart Schilperoort (MSc)
for his assistance with the 3D radiation shield. To the Faculty of Geo-Information Science and Earth Observation from University of Twente
5  and the staff from the water laboratory of Delft University of Technology.

*Competing interests.* The authors declare that they have no conflict of interest.



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





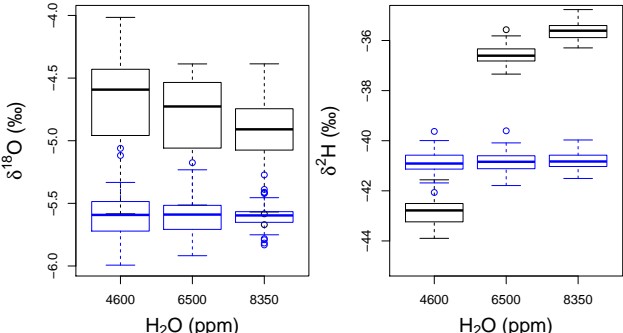

**Figure A1.** Isotope signatures from the raw (black boxplots) and calibrated (blue boxplots) measurements of the known water sample used during the calibration of the experiment 1.

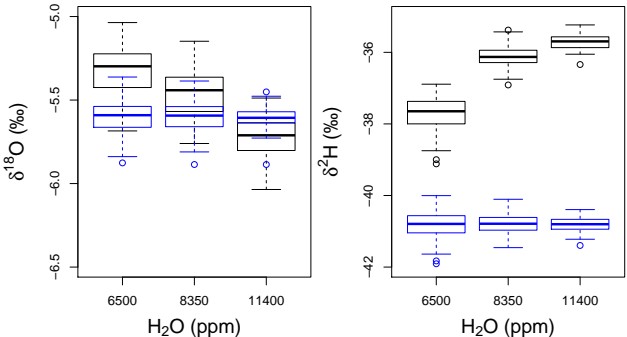

**Figure A2.** Isotope signatures from the raw (black boxplots) and calibrated (blue boxplots) measurements of the known water sample used during the calibration of the experiment 2.

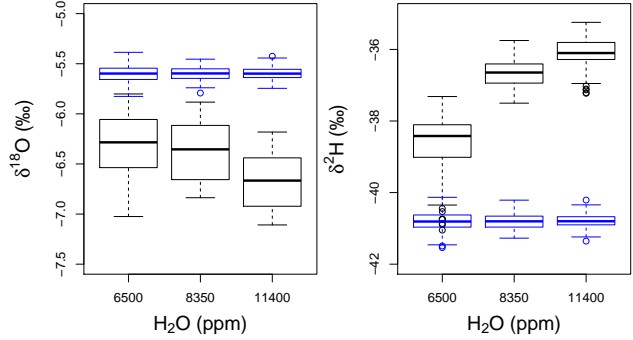

**Figure A3.** Isotope signatures from the raw (black boxplots) and calibrated (blue boxplots) measurements of the known water sample used during the calibration of the experiment 3.