# Peer review of "Technical note: an alternative water vapor sampling technique for stable isotope analysis"

_Hydrology and Earth System Sciences, 2018_

## Referee Comment (RC1) · Anonymous Referee #1 · 13 Nov 2018

General comments

In their manuscript entitled 'An alternative water vapor sampling technique for stable isotope analysis' the authors aim at developing a field vapor sampling method for later water stable isotope analysis of the samples in the laboratory using an OA-ICOS laser-based isotope analyzer (LGR, WVIA 912). They present isotope data from two laboratory experiments and one field sampling campaign. Overall, the manuscript is well structured. However, the language is somewhat imprecise and needs to be improved.

In general, I appreciate the authors' contribution to the field of isotope hydrology and their attempt to establish a water vapor sampling technique for isotope analysis. Since

water vapor sampling for isotope analysis with laser-based analyzers is currently receiving increased attention with several issues still being unsolved, this contribution appears to be of interest for the readers of Hydrology and Earth System Sciences. However, I think that some questions still need to be addressed, as some assumptions are not correct and the conclusions that were drawn thereof are inadmissible. Also many of the results are discussed rather briefly.

One essential point is missing in this study: there is no evidence that one can rely on the air-tightness of the bags. Are the selected bags really appropriate for collecting discrete vapor samples? The magnitude of weight loss of moist samples through the wall of different containers and bag types has been shown previously (Herbstritt et al., 2014). The assumption that the deviation of the laboratory air from the collected air samples is '...good indication that there is no mixing...' is not verified. What would the isotopic composition of the sampled vapor look like after a few days or weeks of storage? Maybe you can't trust the PE bags and the difference in D-Ex between sample A and samples B/C is not due to the different sampling sites one day before but due to evaporation through the wall of the sampling bag? Further, it is not discussed why sample D (= sampled laboratory air) does not match the directly analyzed laboratory air (Table 1). A simple test for the reliability of the sampling bags could be to fill the potential sample containers with dry gas (< 1000 ppm) and check the vapor concentration from time to time over the course of several days or weeks.

Due to the lack of this essential information and the missing proof of the sample bags' reliability, the manuscript cannot be recommended as a reference for operators of this technique and therefore should be rejected.

Specific comments

In my opinion, Experiment 1 and 2 are quite similar to the work of Kurita et al., 2012 and Aemisegger et al., 2012. It is true that response times (Exp. 1) may differ due to the setup, i.e. the dead volume of tubings etc. and due to the between-sample differences

in the isotopic composition (memory effect) but information on the response time is given more or less in the instruments specifications (user manual) already.

Some clarifications of details in the method sections would be necessary, as some important information was skipped e.g. how were the vapor concentrations of the vapor standards generated? Also, "dried" laboratory air with a vapor concentration of around 5000 ppm isn't really "dry"; it rather seems to be quite high compared to the produced vapor standards of 4600 ppm, 6500 ppm and 8350 ppm. Was the produced "dry" air used to dilute the produced vapor standards? Furthermore, in Experiment 2, the memory effect is not reduced by the "dried" air. In this case, it would have been rather reduced by analyzing the samples subsequently, as their isotopic composition is quite similar.

The cold trap samples appear to be enriched in d2H relative to the bag sample data, but in the case of incomplete condensation I would expect that both isotope ratios are affected. This is not the case for d18O. Obviously the cold trap data show higher variabilities than the vapor samples but seem to correspond more or less to the precipitation (liquid) sample at the respective day whereas bag sample data don't. Why is there no difference along the sampled profile in the vapor bags (Exp. 3)? Did the authors expect to see different isotopic compositions in the vertical profile or why was this setup chosen? Would it be possible that the isotope data of the sampled vapor were flawed by diffusive exchange through the bags' wall with ambient air prior to analysis?

References:

Aemisegger, F., Sturm, P., Graf, P., Sodemann, H., Pfahl, S., Knohl, A., and Wernli, H.: Measuring variations of d18O and d2H in atmospheric water vapour using two commercial laser-based spectrometers: an instrument characterisation study, Atmospheric Measurement Techniques, 5, 1491–1511, https://doi.org/10.5194/amt-5-1491-2012, 2012.

Herbstritt, B.; Limprecht, M.; Gralher, B.; Weiler, M. 2014, Effects of soil properties on the apparent water-vapor isotope equilibrium fractionation: Implications for the headspace equilibrium method [poster presentation], UNI Freiburg. Available at http://www.hydro.uni-freiburg.de/publ/pubpics/post229.

Kurita, N., Newman, B. D., Araguas-Araguas, L. J., and Aggarwal, P.: Evaluation of continuous water vapor D and 18O measurements by off-axis integrated cavity output spectroscopy, Atmospheric Measurement Techniques,5, 2069–2080, https://doi.org/10.5194/amt-5-2069-2012, 2012.

---

## Short Comment (SC1) · 13 Nov 2018

General comments:

In their manuscript, Jimenez–Rodriguez et al. describe a method to collect discrete water vapor samples for subsequent analysis on a laser-based water stable isotope analyzer. For this purpose, they investigated in a three-step experiment (a) the minimum air sample volume required for meaningful data acquisition based on the system response time and the ideal aggregation time period, (b) the consistency of isotope data from continuous sampling of unique sources, and (c) the utility of commercially available polyethylene bags for the collection of discrete vapor samples from remote

locations. Conservative collection and storage of large numbers of discrete vapor samples at reasonable costs has been a challenge so far. This is mainly due to the lack of suitable sampling containers that would minimize the risk of samples being compromised during handling and storage due to their small reservoir size and resulting high susceptibility to contamination via exchange with ambient atmospheres. A solution of this issue would significantly expand the potential of the increasingly popular laser-based water stable isotope analyzers which is well within the scope of HESS.

However, there appear to be severe misinterpretations of the presented data. Unfortunately, the authors did not compare their sample bag results with data from alternatives of water vapor stable isotope measurements they would have considered trustworthy. The differences between direct laboratory air measurements and bag-sampled air from the same location (Sample D) are remarkable (Table 1) but ignored in the manuscript. The differences between data from bag-sampled air and cold traps are attributed to the alleged failure of the latter. But then why are the authors showing these data? Comparison of vapor concentrations during sampling and during measurements would have been helpful but are missing.

Specifically, I would have expected the cold trap data to follow a trend line, similar to an evaporation line, in dual isotope space as a result of the alleged incomplete vapor sampling. This was not the case (Fig. 8). Neither did they plot towards the upper right of the sample bag data as must be the case after enrichment in 18O and 2H taking the allegedly unflawed sample bag data as the origin of this evolution. In my perception, the cold trap data may indeed represent the natural variability of sampled air masses. Sample bag isotope data were quite consistent and, moreover, strongly deviating from the cold trap cluster in dual isotope space. However, even if cold trap data were flawed there is no proof that bag samples were not subject to exchange with each other and or via the ambient atmosphere. Conversely, unintended exchange would well explain the similarity of their vapor isotopic compositions. The statement that the mere difference between isotope signatures of laboratory air vapor and bag-sampled vapor is a "good

indication" that no exchange occurred is not justified. And it is proven wrong when some of the bag samples were supposed to represent the very laboratory air.

Polyethylene bags similar to the ones used in this study have been shown to allow for evaporative loss of water resulting in measurable changes of the contained water vapor stable isotopic composition within several days of storage (Hendry et al., 2015, doi: 10.5194/hess-19-4427-2015). This happened despite the enclosed water vapor being in isothermal equilibrium with a markedly bigger liquid water reservoir present in the co-enclosed natural soil sample. The vapor-only reservoirs investigated in this study were several orders of magnitude smaller than a typical soil sample liquid water reservoir (microliters vs. milliliters) and must therefore be expected to reveal measurable changes in their isotopic compositions within mere minutes. This is the reason why commercially available gas sampling bags, e.g. Lindebags, include one layer of diffusion-tight metal foil.

In summary, the authors did not demonstrate that sample bag data do in fact represent what they are claimed to represent. Furthermore, I do not see how a re-interpretation of the presented data would suffice the aim of a reliable method for collecting representative discrete water vapor samples. The first two steps of the described experiment are mainly a repetition of the work of Aemisegger et al. (2012, DOI: 10.5194/amt-5-1491-2012) with insufficient novelty to justify their publication. I therefore regret to say that I recommend rejecting this manuscript. Further, I provide a list of detailed comments below that should help to improve a future manuscript.

—

Specific comments and Technical corrections:

Title: alternative to what?

P1-L4: insert "isotopic" before "fractionation"

P1-L5: the quality of the measurement should be characterized because the analyzer

will provide continuous data regardless of source. However, only after sufficiently long analysis of a sufficiently large reservoir these data will be e.g. representative, stable, reliable, meaningful, or reasonable. + delete "one" + capacity of what?

P1-L7: I know "under . . . conditions" but not "under. . .set up". Please rephrase.

P1-L8: tense: can -> could, allows -> allowed

P1-L11: "resolution", not "variation"

P1-L11: insert "with" before "the cold traps"

P1-L13: given the following sentences, this must be evapotranspiration, not evaporation. What are the provided references referring to?

P1-L16: rephrase to e.g. ". . .surfaces. Their partitioning is. . ." or "... surfaces with their partitioning being ..."

P1-L22: incorrect isotope terminology: delta values do not refer to isotopes but to isotope ratios. Please rephrase.

P2-L1: please be more specific: It's the isotope fractionation factors that depend on temperature.

P2-L1f: rephrase to e.g.: "Physical isotope fractionation is driven by water phase change and also to a lower extent by diffusion." Mixing is a conservative process and does not cause fractionation although it does in fact produce a different isotopic composition in the case of two distinct reservoirs being mixed.

P2-L4: delete "whilst" or connect the two sentences

P2-L5: "caused by", not "caused during"

P2-L7: "unidirectional"? E.g. net evaporation or net condensation is the result of a mismatch between the absolute evaporation flux and the absolute condensation flux. This makes it highly bidirectional.

**HESSD**

P2-L20: start new sentence: "However,..."

P2-L30: please rephrase: the risk is not ISOTOPIC fractionation. This is in fact taken into account. The risk is incomplete sampling.

P3-L1: please quantify "short"

P3-L2: improvements regarding what?

P3-L6: "inTO the field"

P3-L7: what are "controlled run temperatures"? + insert "apply" after "restrictions"

P3-L8: insert "isotope" before "fractionation"

P3-L11: insert "isotope" before "fractionation"

P3-L12: signature -> signatures

P3-L13: insert "the" before "cold"

P3-L16: please be more specific, e.g. "A LGR (ABB - Los Gatos Research Inc., San Jose, CA, USA) Water Vapor..."

P3-L16: signature -> signatures

P3-L18: colloquial language, please rephrase, e.g. "with measurements of liquid water standards of known isotopic composition..."

P3-L22f: I do not understand this sentence, please rephrase

P3-L24: 5000ppm? Dried air should have no more than a few hundred ppm remaining vapor mixing ratio. Please comment on the high number you encountered

P3-L24f: I do not understand this sentence, please rephrase

P3-L30: please specify what makes these standard deviations meaningful. For example, are they sufficient to discriminate samples that represent the natural variation of

isotope ratios on typical timescales?

P4-L1: signature -> signatures

P4-L3: analyses, not analysis + what kind of analyses? + tests, not test

P4-L6: isotope signatures are expressed in delta values, not just in heavy isotopes + "are" or "were" before "expressed"

P4-L7f: please describe the calibration procedure and the correction – if necessary – of drift and vapor concentration effects during liquid water analyses

P4-L8: please define the abbreviation "IWA". Is this the "WVIA"?

P4-L15 (and throughout the manuscript): this is a correction, not a calibration, usually resulting in a normalization of raw isotope data to a reasonable water vapor mixing ratio (see e.g. Schmidt et al., 2010, DOI: 10.1002/rcm.4813 or Johnson et al., 2011, DOI: 10.1002/rcm.4894 for more details). Please state why you chose to do differently

P4-L19: please provide more details on "automatically" + "a" means only one. How many different waters were used for this step?

P4-L21: I do not understand "water molecule concentrations depending on the air sample concentrations"

P4-L22: is -> was

P4-L24: are -> were

P4-L24f: add here that these were calculated using equations 2 & 3 + I suggest to not use the symbol alpha as it represents fractionation rather than correction factors in isotope contexts

P4-L27: value -> values + is -> were

P5-L8: was, not were

P5-L12: "run" -> "was run" or better "was conducted"

P5-L16f: this statement should be placed after the description how one minute was determined as ideal aggregation time period.

P5-L18: insert "statistical" or equivalent before "analysis"

P5-L19: isn't it the standard deviation of the moving average that is governed (not driven – sloppy jargon)? + why not connect the two sentences with "and" as they both start with "This analysis"?

P5-L23: insert "it" before "is"

P5-L25: the Allan deviation plots I know have minima at the respective aggregation time. + Are you referring to Figure 5 here? If so, please state. Furthermore, this figure and its discussion should appear first, as your first decision (i.e. the 1-min aggregation time) is based on it.

P6-Figure 1: I do not understand why this effort was necessary. Why wasn't it sufficient to (perform and) look at the 600 s interval to retrieve the desired information? And isn't this information already provided in Aemisegger et al. (2012, DOI: 10.5194/amt-5-1491-2012) or could have been concluded from the injection frequency and valve operation pattern of routine liquid water analyses performed on such analyzers? Similar objections apply for the aggregation time.

P6-L5: why were polyethylene bags selected despite being aware of the findings of Hendry et al. (2015, doi: 10.5194/hess-19-4427-2015)? See general comments for details.

P6-L6ff: such paragraphs should be written in past tense

P6-L11: it has -> with

P6-L12: insert "for" before "the tight" + what do you mean by "movement"?

P7-L1: "Air samples were collected manually..."

P7-L6: why didn't you normalize all measurements to e.g. 10k ppm i.e. calculate the raw isotope numbers the analyzer would have shown if the vapor concentration had been 10k ppm (see e.g. Schmidt et al., 2010, DOI: 10.1002/rcm.4813 or Johnson et al., 2011, DOI: 10.1002/rcm.4894 for details), prior to calibration?

P7-L8: statistical -> statistically significant

P7-L11: commonly, the deuterium excess is indicated by the lower case letter d (in italics)

P8-L7: an -> a

P8-L11: were -> was

P8-L13: it would be important to read that the tubes reached the bottom of the bottles in a way that only the minimized inner cross section area of the tubes allowed for the interfacial exchange between sampled water and ambient atmosphere. Were they installed that way? 15 cm sounds a little short for 5-L bottles. And 9 mm sounds a little wide for this purpose. How would this affect your LMWL?

P8-L14: 6mm inner or outer diameter? Both of which appear quite a lot. + "reduce the vapour exchange" -> "facilitate pressure compensation while at the same time minimizing loss via vapor diffusion" or equivalent. Pressure compensation is necessary once the inlet tube is submersed into the sampled water which should happen as soon as possible (see previous comment)

P8-L18: simultaneously to the cold trap or to each other?

P8-L19: why not 24? (4 h * 6 samples/h = 24 samples)

P8-L21: please rephrase and start new sentence (the frozen vapor was not closed nor did it collect...), e.g. "The liquid water sample were immediately transferred into..."

[Figure]

P8-L23: delete "a"

P8-L25: "the" not "its", because vapor was measured, not vapor condensation or sampling bags

P8-L25f: I do not understand this statement. Wasn't the concentration at which the samples were analyzed just the one present inside the bags?

P9-Figure 3: this figure should appear in the method section. Throughout the manuscript, all figures should appear near their description.

P9-L8: insert "probably" before "because", as this is your speculation

P9-L11: please start new sentence ("However,...") + I do not understand "some averages with non stable measurements"

P9-L12: please make sure that figures and their description and discussion appear in the right order

P10-Figure 4: see comment on P6-Figure 1

P10-Figure 5: I am unable to find 0.3‰ or 1.5‰ on the vertical axis, thus I am unable to see what aggregation time is sufficient to reach these standard deviations + the numbers on the vertical axis are not evenly spaced + the label of the horizontal axis should be "aggregation time" or equivalent + this figure and its discussion should appear before any figure featuring 1-min-means because those were chosen based on this analysis of the Allan deviation + "moving", not "mobile"

P10-L1: aren't 450 mL calculated quite tightly? What if you have two strongly differing successive samples and the memory effect causes the readings from the second sample to not have stabilized after 240s leaving not enough time for a 1-min-average before the bag is empty? Further, the smaller the vapor reservoir, the higher its susceptibility to contamination + delete "to carry out 300 s of continuous measurements" as you provided this number in the previous sentence already

P11-L2: I do not understand why this step was necessary. Do you have indication that small vapor reservoirs such as your sampling bags would reveal significant variations? If so please elaborate on this also in the introduction

P11-L4: insert "probably" before "because", as this is your speculation

P11-L6: whose capacity?

P11-L8: signature -> signatures + insert "those of" after "from"

P11-L8f: I strongly disagree with this statement. From your experimental design and data, there is no way of telling whether your bag samples are or are not a mixture of the original (e.g. flux tower) sample and other sources. This would only have been possible if you had analyzed a distinct air directly and sampled it into bags in parallel, then stored the bags while exposing them to a different ambient atmosphere, then analyzed the bag air, and then compared the results of direct and discrete sample measurements.

P11-L10: statistical -> statistically significant + insert "and" between the two delta expressions

P11-L11: insert "probably" before "the reason", as this is your speculation

P11-L12: insert "probably" before "because", as this is your speculation + are you referring to the absolute deviation of your arithmetic mean from the true value (i.e. the accuracy) or are you rather referring to the standard deviation (i.e. the precision)? + on -> of

P11-L13: showed -> observed

P11-L14: is this the within-sample or the between-sample deviation?

P11-L15: why would wind change the isotopic composition? Are you referring to different air parcels being sampled?

[Figure]

P11-L16: sampling -> sample + per set of sample or per sample?

P12-Table 1: shouldn't laboratory air and sample D be consistent? Could it be that the sample bags were stored in a confined space where they exchanged with each other? The consistency among A-D is striking. And so is the discrepancy between laboratory air and sample D. Why was sample D not discussed in the manuscript? This could have been an indication that bag samples represent what they are supposed to represent. In order for a potential consistency of lab air and D to be a proof, conditions as described in comment to P11-L8f would have been necessary + in the figure caption: lower case -> superscript + this needs more details. What exactly is different when a, b, c, or d is displayed?

P12-L4: delete "or after" or write "or later on that day"

P12-L5f: given, that you report the LGR measurements in ppm, can you provide ppm values for the observed humidity as well? This might give the reader a clue whether your sampling was conservative or exchange with ambient air has occurred

P12-L10: the offset of the equation has the "unit" ‰

P12-L11: is located -> plots + "heavier" is too colloquial, please rephrase

P12-L12: insert "isotopic signatures" before "water vapor

P12-L13: "lighter" is too colloquial, please rephrase + insert delta symbols before 2H and 18O

P12-L14: insert "show" before "less variation"

P12-L14f: but shouldn't it be the opposite? Cold trap samples should represent a mixture of six potentially variable bag samples. The similarity among air samples leads me to the conclusion that the originally present natural variation, still revealed to some degree by the cold trap data, got completely lost when all bag samples exchanged with or via a similar atmosphere prior to analysis
P12-L15f: the location of atmospheric vapor isotope signatures relative to the LMWL also depend on the slope thereof

P12-L17f: aren't these interpretations referring to liquid water? You are showing vapor data. Therefore, you first have to determine where the corresponding liquid water reservoir would plot relative to the LMWL before making these statements

P12-L21: insert "isotope" before "signature"

P12-L22: heavier delta2H -> higher delta2h values + incomplete at -70°C? What was the remaining vapor pressure at the cold trap outlet? Assuming that cold trap data might be flawed, why did you present them as a reference for the sample bag data? Why was it not possible to design the experiment in a way that the reference data set (i.e. cold trap) is trustworthy? Further, wouldn't incomplete condensation result in a trend line extending to the upper right of the sample bag data rather than in a data cloud located towards the upper left? + enrichment of what?

P12-L23: delete "it" + replace "is" by "may be", as this is your speculation and strongly depends on setup properties

P13-Figure 7: I suggest "time of day (hh:mm)" as label of the horizontal axis

P13-L3: all -> the entire + exception from big differences? + is shown -> was observed

P13-L4: "Isotope signature at 34m height from cold traps..." -> "Isotope signatures of samples collected at 34 m height via cold traps..."

P13-L4f: what if these data were the true numbers and your bag sample data were flawed?

P13-L7: enriched in what?

P13-L12: what was the air flow rate through the cold traps? This information would be useful to calculate the minimum humidity (in ppm) during sampling. Based on this estimate, further interpretations of the sample bags' diffusion-tightness might be possible

P14-L7: insert "it" after "experiment 2" and "isotope" after "stable"

P14-L7f: please rephrase this sentence

P14-L10: resolution, not variation + please connect the two sentences or rephrase

P14-L12: please connect the two sentences or rephrase

P14-L14: dew temperature -> dew point

P14-L15: of -> or + insert "the dew point in" before "the field". Do you have a suggestion how to deal with this situation?

P15-Figure 9: proportional to -> indicating

---

## Referee Comment (RC2) · Anonymous Referee #2 · 23 Nov 2018

The technical note "An alternative water vapor sampling technique for stable isotope analysis" by Jiménez-Rodriguez et al. presents a method to sample water vapor for posterior analysis with a laser-based isotope analyser. To validate the proposed methodology, the authors present 3 experiments: (i) A laboratory experiment to test response time of the proposed system; (ii) A laboratory experiment to test the consistence of the sampling methodology, and (iii) a field experiment comparing the proposed methodology with the cold trap sampling procedure. Since finding a methodology to sample water vapor for isotope analysis in remote locations is an important challenge, I consider that the contribution is of great interest for the readers of Hydrology and Earth System Sciences. However, in my opinion there are some important aspects that are

not clearly justified, and as a consequence the conclusions are based in weak foundations. In my opinion, the two main important aspects not justified are: 1) The isotopic differences between the lab air, sampled directly, and lab air sampled with the bags are significant. Table 1 presents larger differences between the lab air directly sampled (Laboratory) and sampled with the bags (sample D), than differences between bag sampled laboratory air (sample D) and forest air (sample A). These differences are not explained, nor justified in the manuscript. 2) The field experiment comparing the isotopic composition of the air sampled with the bags and the air sampling with the cold trap method, gives important differences between methods. Then, authors conclude that differences are due to inappropriate results given by the traditional cryogenic collection technique, compared to results given by the method proposed. However, there is not clear justification of this conclusion in the manuscript. In addition, there are several statements that are not clearly justified. (for example: lines 7-8 in page 11; lines 17-19 in page 12; lines 4-5 in page 13; lines 7-10 in page 14). For that reasons, the manuscript cannot be recommended to be published in HESS.

---

## Author Comment (AC1) · 15 Jan 2019

We would like to thank the reviewer for his/her valuable comments on our manuscript. We appreciate that the reviewer acknowledges the interest and relevance of our study. Nonetheless, the reviewer indicated some issues which we will clarify point-by-point:

1) One essential point is missing in this study: there is no evidence that one can rely on the air-tightness of the bags. Are the selected bags really appropriate for collecting discrete vapor samples? The magnitude of weight loss of moist samples through the wall of different containers and bag types has been shown previously (Herbstritt et al., 2014). The assumption that the deviation of the laboratory air from the collected air

samples is '. . .good indication that there is no mixing. . .' is not verified. What would the isotopic composition of the sampled vapor look like after a few days or weeks of storage? Maybe you can't trust the PE bags and the difference in D-Ex between sample A and samples B/C is not due to the different sampling sites one day before but due to evaporation through the wall of the sampling bag?

Reply: We agree that this was missing, so following the comments and recommendations from both reviewers and Mr. Gralher, we developed an extra experiment aiming to clarify the performance of the LDPE sampling bags. The full experiment is available as an appendix to this reply and as a proposal to add its results to this manuscript. In this extra experiment we compared the consistency of air stored in our LDPE-bag to two commercial air-sampling bags: Tedlar and Foil bags. We collected air samples and stored them in these different types of bags on 'day 0' and analysed the samples 1,2,9, 16 and 17 days after collection. The bag samples were also compared to direct measurements of the laboratory air by the WVIA. Additionally, we also weighted the bags to check the conservation of mass.

As the main result, we confirm the low reliability of the LDPE bags when compared to the foil bags and direct measurements with the WVIA. However, the performance of the LDPE is better than the performance of the cold traps. We also found that not only our LDPE bag was affected by the ambient laboratory air, but also the other 2 bags likely due to the water vapor transmission rate of the different materials (Tock, 1983). This confirms the pattern described by Herbstritt et al. (2014) who depicted the effect of water mass loss by diffusion through the wall of different sampling materials for the determination of stable isotope signatures of soil pore water under equilibrium conditions. This experiment and its results are added as supplemental material to the reply to the reviewers.

2) Further, it is not discussed why sample D (= sampled laboratory air) does not match the directly analyzed laboratory air (Table 1). A simple test for the reliability of the sampling bags could be to fill the potential sample containers with dry gas (< 1000

ppm) and check the vapor concentration from time to time over the course of several days or weeks.

Reply: Thank you for pointing out this issue. In fact, this interpretation is a misunderstanding from our methods where it was not clearly explained that the sampling date in the laboratory is not the same as when the samples were analysed. Hence 'sample D' and 'laboratory' should not have a similar isotopic value per se. In method section 2.2.4, page 7, line 3 we omitted to mention that Sample D was collected one week before the measurement at the laboratory. We added the "laboratory" sample as a background validation during analysis to show the "Laboratory" as a proof of that no laboratory air was leaking and/or mixing within the inlet used to convey the air from the sampling bag to the Multiport Inlet Unit. We agreed that we haven't mentioned the time difference between the sample collection and sample analysis.

Due to the lack of this essential information and the missing proof of the sample bags' reliability, the manuscript cannot be recommended as a reference for operators of this technique and therefore should be rejected.

Reply to Specific Comments

1) In my opinion, Experiment 1 and 2 are quite similar to the work of Kurita et al., 2012 and Aemisegger et al., 2012. It is true that response times (Exp. 1) may differ due to the setup, i.e. the dead volume of tubings etc. and due to the between-sample differences in the isotopic composition (memory effect) but information on the response time is given more or less in the instruments specifications (user manual) already.

Replay: The aim of experiment 1 was set as "The first experiment (Section 2.2.3) aimed to determine the minimum air volume required to obtain measurements of 2H and 18O with standard deviations lower than 1.5‰ and 0.30‰ respectively to ensure reliable measurements (Kurita et al., 2012)." We cited Kurita et al. (2012) as the source of our target accuracy for individual samples, whilst the methodology of the experiment 1 aims to describe the time required for the specific conditions of our device.

This is important for all users in order to define the specific time required to get a stable signature depending on the tubing length of their laboratory setup, as it is mentioned later on by the same reviewer: "It is true that response times (Exp. 1) may differ due to the setup, i.e. the dead volume of tubings etc. and due to the between-sample differences in the isotopic composition (memory effect) but information on the response time is given more or less in the instruments specifications (user manual) already". Therefore, we decided to provide a practical way to determine the required volume considering the specific device and setup used.

In respect to Aemisegger et al. (2012), we state in Section 2.2.3 that we are following the Aemisegger et al.'s methodology. This one was applied to check whether the minimum volume defined in the previous section of the same experiment is driven by the device's white noise or not: "This analysis determines the time interval from where the moving average is driven by the laser white noise and not by the memory effect of the previous sample. This analysis evaluates the difference between consecutive measurements (yi and yi+1) aggregated at the same time interval ($\tau$) and averaged over the total number of measurements (n) (Aemisegger et al., 2012; Allan, 1966)".

2) Some clarifications of details in the method sections would be necessary, as some important information was skipped e.g. how were the vapor concentrations of the vapor standards generated?

Replay: In Section 2.1 Instrumentation it states that: "A Water Vapor Isotope Analyzer (WVIA; model 912) was used to determine the isotope signature of water vapor samples, with the support of a LGR Water Vapor Isotope Standard Source (WVISS; model 908-0004-9002) employed to provide a controllable flow of water vapor with a known liquid standard measurement for an absolute calibration of raw measurements". Additionally, the methodology of each experiment describes the specific water vapor concentrations used for the calibration.

3) Also, "dried" laboratory air with a vapor concentration of around 5000 ppm isn't really

"dry"; it rather seems to be quite high compared to the produced vapor standards of 4600 ppm, 6500 ppm and 8350 ppm. Was the produced "dry" air used to dilute the produced vapor standards?

Replay: Actually, we used the term "dried" instead of dry because we did not completely dry the air on purpose. In the Section 2.2.3 Experiment 1: Response Time, we mention the following: "In order to identify the memory effect from a previous analysis, ambient air from the laboratory and dried air were measured alternately with the same time intervals (Fig.1)". This shows that the idea of alternating the target sample with dried air from the laboratory was to provide a completely different isotope signature before the target sample. In this way, we were able to determine the precise moment when the signature recorded by the WVIA has no memory effect anymore of the previous sample. Additionally, the "dried air" collected with the MIU is different from dry air required for the WVISS.

Consequently, we propose to add the following sentence on page 3, line 21 (Section 2.1) to clarify the difference between the dried air for the MIU and the dry air source for the WVISS:

"The dry air needed for the WVISS was provided by the Dry Air Source (DAS) device from LGR."

We hope that this will clarify that the "dried air" used on the first inlet of the MIU is completely different than the dry air required for the dilution of the water standard source (DAS).

4) Furthermore, in Experiment 2, the memory effect is not reduced by the "dried" air. In this case, it would have been rather reduced by analyzing the samples subsequently, as their isotopic composition is quite similar.

Replay: This is a misunderstanding of the following sentence: "The variation provided by the source of dried air in between samples allows to identify among individual samples and to observe the capacity to reduce the memory effect from the previous air sample". In this case, we refer to the capacity of the first 2 min of measurements to get rid of the memory effect from the isotopic signature provided by the dried air.

5) The cold trap samples appear to be enriched in d2H relative to the bag sample data, but in the case of incomplete condensation, I would expect that both isotope ratios are affected. This is not the case for d18O. Obviously, the cold trap data show higher variabilities than the vapor samples but seem to correspond more or less to the precipitation (liquid) sample at the respective day whereas bag sample data don't.

Replay: According to the results from the additional experiment performed, we decided to remove this experiment from the manuscript.

6) Why is there no difference along the sampled profile in the vapor bags (Exp. 3)? Did the authors expect to see different isotopic compositions in the vertical profile or why was this setup chosen? Would it be possible that the isotope data of the sampled vapor were flawed by diffusive exchange through the bags' wall with ambient air prior to analysis?

Replay: We were expecting to see differences among heights as a consequence of transpiration from the canopy. After the additional experiment we carried out, we agree with the reviewer about the diffusive exchange through the bag's wall. However, the additional experiment showed that cold traps differ strongly from the direct measurements with the WVIA and the error between cold traps and benchmark are bigger than the LDPE bags.

References

Aemisegger, F., Sturm, P., Graf, P., Sodemann, H., Pfahl, S., Knohl, A., and Wernli, H. Measuring variations of d18O and d2H in atmospheric water vapour using two commercial laser-based spectrometers: an instrument characterisation study, Atmospheric Measurement Techniques, 5, 1491–1511, https://doi.org/10.5194/amt-5-1491-

HESSD

2012, 2012.

Allan, D.: Statistics of atomic frequency standards, Proceedings of the IEEE, 54, 221–230, https://doi.org/10.1109/PROC.1966.4634, 1966.

Herbstritt, B., Limprecht, M., Gralher, B., Weiler, M.: Effects of soil properties on the apparent water-vapor isotope equilibrium fractionation: Implications for the headspace equilibrium method [poster presentation], UNI Freiburg. Available at: http://www.hydro.uni-freiburg.de/publ/pubpics/post229, 2014.

Kurita, N., Newman, B. D., Araguas-Araguas, L. J., and Aggarwal, P.: Evaluation of continuous water vapor D and 18O measurements by off-axis integrated cavity output spectroscopy, Atmospheric Measurement Techniques,5, 2069–2080. , https://doi.org/10.5194/amt-5-2069-2012, 2012

Orlowski, N., Pratt, D. L., and McDonnell, J. J.: Intercomparison of soil pore water extraction methods for stable isotope analysis. Hydrol. Process., 30: 3434–3449. , https://doi.org/10.1002/hyp.10870, 2016

Tock, R. W.: Permeabilities and water vapor transmission rates for commercial polymer films. Advances in Polymer Technology: Journal of the Polymer Processing Institute, 3(3), 223-231. https://doi.org/10.1002/adv.1983.060030304, 1983.

Wassenaar, L. I., Ahmad, M., Aggarwal, P., Duren, M., Pöltenstein, L., Araguas, L. and Kurttas, T.: Worldwide proficiency test for routine analysis of $\delta$2H and $\delta$18O in water by isotope‐ratio mass spectrometry and laser absorption spectroscopy. Rapid Commun. Mass Spectrom., 26: 1641-1648. , https://doi.org/10.1002/rcm.6270, 2012.

Please also note the supplement to this comment: https://www.hydrol-earth-syst-sci-discuss.net/hess-2018-538/hess-2018-538-AC1-supplement.pdf

538, 2018.

**Supplement:**

**Supplemental Material: Additional Experiment**

**Objective**

Determine the capacity to retrieve similar isotope signatures of air samples collected with different sampling methods for water vapor samples.

**Methodology**

On the market different sample bags exist to store air samples. In this experiment we test whether these stored air samples remain isotopically consistent in time. We tested the isotopic signature of the air samples stored for a period of 17 days. We tested 3 different types of bags that we all collected on the same day ($T_0$). Successively we analysed them on the same day ($T_0$), and after 1, 2, 9, 16, and 17 days after collection ($T_1$, $T_2$, $T_9$, $T_{16}$, $T_{17}$, respectively). We compared the results to direct air vapour sampling with the open inlets from the MIU unit on day $T_0$, this measurement is our 'benchmark'. Additionally, we also collected liquid samples with two cold traps with different pumping rates on day $T_0$. One cold trap experiment was carried out with a fast pumping rate (FPR) and with a slow pumping rate (SPR).

The three types of bags tested are:

- MPE bags: these 1 L bags of methalized polyethylene are manufactured with a five layer structure (Foil Bag; Model: FP-1, Samplebags.eu) and designed to be filled to 90% of its volume capacity. Every bag has a 2-in-1 PTFE fitting for the injection and extraction of the air sample.
- PVF bags: these 1 L bags are composed of polyvinyl fluoride (Tedlar Bag; Model: ITP-1, Samplebags.eu) and designed to be filled to 90% of its volume capacity. Every bag has a 2-in-1 PTFE fitting for the injection and extraction of the air sample.
- LDPE bags: these 1.1 L bags are made of low density polyethylene used for filling packaging spaces. The sampling bags are fabricated with a simple valve made from polyethylene as well.

Both, MPE and PVF bags are designed to be filled to 90 % of its volume capacity and every bag has a 2-in-1 PTFE fitting for the injection and extraction of the air sample. Whilst the LDPE bags are fabricated with a valve made from polyethylene as well. The collection of the samples took 3 hours. Each hour we filled 6 sample bags per bag type and carried out the cryogenic sampling with two cold traps. The cryogenic samples were collected through cold traps built with two different pumping rates: 50 mL min$^{-1}$ (slow pumping rate) and 3 L min$^{-1}$ (fast pumping rate). The cold traps were built with a test tube of 50 mL capacity immerse in a container filled with ethanol (100 %) inside a cooler filled with dry ice (-70 ºC). The water collected in both test tubes was thaw and transferred to a 1.5 mL vial for its measurement after the experiment with the LWIA. The cryogenic sampling was performed during the 3 hours of sampling, collecting 3 samples with the FPR and only one with the SPR. We also took our benchmark sample by using the open inlet of the WVIA, collecting 8 samples per hour during the 3 hours experiment (24 samples). During the analysis of the sample bags on days $T_1$, $T_2$, $T_9$, $T_{16}$ and $T_{17}$, we also sampled the laboratory air with one open inlet of the MIU. The samples were analysed and calibrated according to the procedure as described in Section 2.2.1 and Section 2.2.2, using an standard water with an isotope signature of $\delta^{18}$O: -14.44 ‰ and $\delta^2$H: -104.09 ‰.

The consistency analysis of the isotopic signatures was performed comparing the isotope signatures obtained from the different samples against the benchmark collected the same day of the sampling ($T_0$), which corresponds to the direct measurements from the WVIA. The cross comparison was performed with the *Z* analysis (Equation 1) (Orlowski *et al.* 2016, Wassenaar *et al.* 2012). Where *S* is the isotope signature ($\delta^2$H or $\delta^{18}$O) of the bags or cryogenic samples, *B* is the benchmark isotope signature (WVIA) and $\mu$ is the target variability. Differing from Orlowski *et al.* (2016) and Wassenaar *et al.* (2012), the target variability ($\mu$) was stablished as the isotope range measured with the WVIA during the three hours of measurements of the benchmark ($\delta^2$H: 2.0 ‰ and $\delta^{18}$O: 0.4 ‰) considering the transient condition in the laboratory. Thus, we adopted the limits proposed by Orlowski *et al.* (2016) for accurate analysis (*Z-score* < 2.0), questionable analysis (Z-score: 2.0 – 5.0) and unacceptable analysis (Z-score: > 5.0).

$$Z = \frac{S - B}{\mu} \qquad\qquad\qquad \textit{Equation 1}$$

[Figure]

Figure 1. Air sampling bags used to test the consistency among air sampling methods. Silver bag on the left is the MPE bag, the middle transparent bag corresponds to the PVF bag and the transparent bag on the right is the LDPE bags.

**3. Results and Discussion**

Benchmark isotope signature from the 24 measurements performed during the three hours experiment had an isotope signature of -15.61 ± 0.14 ‰ and -115.12 ± 0.47 ‰ for $\delta^{18}O$ and $\delta^2H$, respectively. All the vapor samples collected with the bags that were measured on the same sampling day ($T_0$) are located within the accurate region based on the *Z-score* analysis (Figure 2). Contrary to these samples, the liquid samples collected with the cold traps are located on the unacceptable region, showing *Z* values bigger than 5. The SPR is only based on one sample because we were only able to collect 0.1 mL of liquid during the three hours, while the FPR collected 0.25 mL every hour. Both liquid samples show a heavier signature than the benchmark, as a consequence of incomplete condensation. These differences are linked to not perfect collection efficiencies during the cryogenic sampling with the cold traps (Griffis, 2013).

The signature of the laboratory air changed during the course of our measurements. The orange samples marked as "Laboratory" in Figure 2 depict the differences among the days of measurement and the sampling date ($T_0$). These differences in laboratory air signatures partly influences the measurement results from all the air samples collected with the three types of bags. The MPE samples are the only ones from the experiment with almost all measurements located within the accurate region of the *Z-score* plot (*Z-score* < 2). Despite the accuracy reached with the MPE, the measurements are influenced by the isotope signature of the air within the laboratory. All the measurements after the sampling date with the LDPE and PVF bags are located within the questionable region of the Z-score plot (*Z-score:* 2-5), while the PVF samples from $T_9$ are on the unacceptable region (*Z-score* > 5). These sampling bags are influenced by the isotopic signature of the laboratory air considering its location close to the laboratory signature during the measurements.

Herbstritt *et al*. (2014) depicted the effect of water mass loss by diffusion through the wall of different sampling materials for the determination of stable isotope signatures of soil pore water under equilibrium conditions. They mention the capacity of the layered foil bags (similar material to the MPE bags used in this experiment) to prevent the vapor mass loss during long periods of time. However, the sampling bags design

influences strongly the capacity to retain stable samples. Tock (1983) evaluates the water vapor transmission rate (WVTR) of different polymers films, finding that methalized polyethylene coated materials have the lowest WVTR (1.55 g $m^{-2}d^{-1}$) compared with the LDPE (23.25 g $m^{-2}d^{-1}$) and the PVF (50.22 g $m^{-2}d^{-1}$). The WVTR defines the capacity of a film to transfer water molecules depending on the relative humidity gradient (Kumaran, 1998). Thus, the WVTR of each material provides insights about the variation of the stable isotope measurements, including the MPE bags. It is important to remark that the diffusion characteristics of foil layered materials are directly influence by the temperature in outdoor conditions (Pons *et al.*, 2014).

[Figure]

Figure 2. Dual plot for the *Z-score* values of $\delta^2$H and $\delta^{18}$O of the samples under analysis.

**3. Conclusions**

Water vapor sampling techniques differ in their capacity to keep reliable measurements after the sampling. The MPE bags shows the more accurate measurement of stable isotopes two weeks after its sampling. LDPE and PVF sampling bags can be used for sampling water vapor if the measurements are performed on the same day of sampling. After 24 hours, the WVTR of both materials (LDPE and PVF) allows the air sample to be in equilibrium with the surrounding air affecting the measurement accuracy. Both cold trap systems differ strongly from the benchmark in this experiment. Apparently, even the 3 hours sampling was not sufficient for a full condensation. This long sampling time, limits its use for studying evaporation process that change in short periods of time (< 30min). Thus, underline the need to determine with further research if cryogenic samples

can be used as references or benchmarks when those are compared against direct measurements performed with laser spectrometers as the WVIA.

**References**

Griffis, T. J. (2013) Tracing the flow of carbon dioxide and water vapor between the biosphere and atmosphere: A review of optical isotope techniques and their application, Agricultural and Forest Meteorology, 174-175, 85 – 109. doi: 10.1016/j.agrformet.2013.02.009

Herbstritt, B., Limprecht, M., Gralher, B., Weiler, M. (2014) Effects of soil properties on the apparent water-vapor isotope equilibrium fractionation: Implications for the headspace equilibrium method [poster presentation], UNI Freiburg. Available at: http://www.hydro.uni-freiburg.de/publ/pubpics/post229.

Kumaran, M. (1998) Interlaboratory Comparison of the ASTM Standard Test Methods for Water Vapor Transmission of Materials (E 96-95). Journal of Testing and Evaluation, 26: 83-88. https://doi.org/10.1520/JTE11977J.

Orlowski, N., Pratt, D. L., and McDonnell, J. J. (2016) Intercomparison of soil pore water extraction methods for stable isotope analysis. Hydrol. Process., 30: 3434–3449. doi: 10.1002/hyp.10870

Pons, E., Yrieix, B., Heymans, L., Dubelley, F., & Planes, E. (2014). Permeation of water vapor through high performance laminates for VIPs and physical characterization of sorption and diffusion phenomena. Energy and Buildings, 85, 604-616. Doi: 10.1016/j.enbuild.2014.08.032

Tock, R. W. (1983). Permeabilities and water vapor transmission rates for commercial polymer films. Advances in Polymer Technology: Journal of the Polymer Processing Institute, 3(3), 223-231. doi: 10.1002/adv.1983.060030304

Wassenaar, L. I., Ahmad, M., Aggarwal, P., Duren, M., Pöltenstein, L., Araguas, L. and Kurttas, T. (2012), Worldwide proficiency test for routine analysis of $\delta 2H$ and $\delta 18O$ in water by isotope-ratio mass spectrometry and laser absorption spectroscopy. Rapid Commun. Mass Spectrom., 26: 1641-1648. doi:10.1002/rcm.6270

---

## Author Comment (AC2) · 15 Jan 2019

We would like to thank the reviewer for his/her valuable comments on our manuscript. We appreciate that the reviewer acknowledges the interest and relevance of our study. Nonetheless, the reviewer indicated some issues which we will clarify point-by-point:

In my opinion, the two main important aspects not justified are:

1) The isotopic differences between the lab air sampled directly, and lab air sampled with the bags are significant. Table 1 presents larger differences between the lab air directly sampled (Laboratory) and sampled with the bags (sample D), than differences

between bag sampled laboratory air (sample D) and forest air (sample A). These differences are not explained, nor justified in the manuscript.

Reply: Thank you for pointing out this issue. In fact, this interpretation is a misunderstanding from our methods where it was not clearly explained that the sampling date in the laboratory is not the same as when the samples were analyzed. Hence 'sample D' and 'laboratory' should not have a similar isotopic value per se. In method section 2.2.4, page 7, line 3 we omitted to mention that Sample D was collected one week before the measurement at the laboratory. We added the "laboratory" sample as a background validation during analysis to show the "Laboratory" as a proof of that no laboratory air was leaking and/or mixing within the inlet used to convey the air from the sampling bag to the Multiport Inlet Unit. We agreed that we haven't mentioned the time difference between the sample collection and sample analysis.

Consequently with the finding from the additional experiment, we decided to remove this section from the experiment to be replaced with the additional experiment (Supplemental Material).

2) The field experiment comparing the isotopic composition of the air sampled with the bags and the air sampling with the cold trap method gives important differences between methods. Then, authors conclude that differences are due to inappropriate results are given by the traditional cryogenic collection technique, compared to results given by the method proposed. However, there is no clear justification of this conclusion in the manuscript.

Reply: We agree with the reviewer that this comparison is not well explained. Our original aim of Experiment 3 was to 1) compare our method to the traditional cold trap method, and 2) test its application in a field setting. Unfortunately, Aim 1 was not well carried out since the cryogenic extraction was not well carried out. For the cryogenic trap method in the field, we used a pumping rate of 3 L min-1. This high rate was necessary to convey the air from the tower towards the sampling point used during

the sample collection (page 8, line 8-9: "... sucking the air at a rate of 3 L min-1 for a travel time of less than 2 min from the sampling point to the collection point."). The problem of this high pumping rate is that it does not allow for a full condensation since the time in the cold bath is too short. Hence the large variability of the cold tap samples and the deviation between the cold trap samples and our method is likely due to this issue. However, after we carried out the additional experiment added as supplementary material to this reply we show the strong differences between the cold trap samples and the LDPE samples, MPE samples, PVF samples and the benchmark (direct measurements from the WVIA). Based on these results we decided to shift the scope of the paper and delete experiment 3 from the manuscript since the LDPE-samples and the cold trap were not reliable measurements.

Specific comments:

In addition, there are several statements that are not clearly justified. (for example: lines 7-8 in page 11; lines 17-19 in page 12; lines 4-5 in page 13; lines 7-10 in page 14). For that reasons, the manuscript cannot be recommended to be published in HESS.

Reply: Page 11, lines 7-8 were clarified on the first paragraph (point 1) of this reply.

Page 12, lines 17-19 says: "Sampling bag signatures depict an isotope signature slightly evaporated. This mixture of vapor could be originated from water evaporated from intercepted surfaces and transpired water from different sources than previous rain events." This sentence aims to provide a possible explanation about the source of the water. However, as the paper's objective is about the applicability of this method we propose to remove both sentences from the paper.

Page 14, lines 7-10 say "Thus, because of the homogeneous mixing within the sampling bags allowing isolating the sample from the surrounding air, preventing contamination and/or mixing during transport and analysis. The homogeneity reached by the air sample within the sampling bag allowed retrieving a better temporal variation than

cold traps during experiment 3."

The homogeneous mixing within the sampling bags is mentioned on page 11, lines 2-3: "Air samples collected with the 1.1 L bags were able to provide a stable signature after two minutes of analysis (Fig. 6)". However, after the performance of the additional experiment added as a supplementary material to this reply we agree with the reviewer that the LDPE bags do not provide a full protection for contamination as a consequence of the bag's diffusivity for water vapor as it is explained by Tock (1983) and later on mentioned by Herbstritt et al. (2014.).

References

Herbstritt, B., Limprecht, M., Gralher, B., Weiler, M.: Effects of soil properties on the apparent water-vapor isotope equilibrium fractionation: Implications for the headspace equilibrium method [poster presentation], UNI Freiburg. Available at: http://www.hydro.uni-freiburg.de/publ/pubpics/post229, 2014.

Tock, R. W.: Permeabilities and water vapor transmission rates for commercial polymer films. Advances in Polymer Technology: Journal of the Polymer Processing Institute, 3(3), 223-231. https://doi.org/10.1002/adv.1983.060030304, 1983.

Please also note the supplement to this comment:
https://www.hydrol-earth-syst-sci-discuss.net/hess-2018-538/hess-2018-538-AC2-supplement.pdf

**Supplement:**

**Supplemental Material: Additional Experiment**

**Objective**

Determine the capacity to retrieve similar isotope signatures of air samples collected with different sampling methods for water vapor samples.

**Methodology**

On the market different sample bags exist to store air samples. In this experiment we test whether these stored air samples remain isotopically consistent in time. We tested the isotopic signature of the air samples stored for a period of 17 days. We tested 3 different types of bags that we all collected on the same day ($T_0$). Successively we analysed them on the same day ($T_0$), and after 1, 2, 9, 16, and 17 days after collection ($T_1$, $T_2$, $T_9$, $T_{16}$, $T_{17}$, respectively). We compared the results to direct air vapour sampling with the open inlets from the MIU unit on day $T_0$, this measurement is our 'benchmark'. Additionally, we also collected liquid samples with two cold traps with different pumping rates on day $T_0$. One cold trap experiment was carried out with a fast pumping rate (FPR) and with a slow pumping rate (SPR).

The three types of bags tested are:

- MPE bags: these 1 L bags of methalized polyethylene are manufactured with a five layer structure (Foil Bag; Model: FP-1, Samplebags.eu) and designed to be filled to 90% of its volume capacity. Every bag has a 2-in-1 PTFE fitting for the injection and extraction of the air sample.
- PVF bags: these 1 L bags are composed of polyvinyl fluoride (Tedlar Bag; Model: ITP-1, Samplebags.eu) and designed to be filled to 90% of its volume capacity. Every bag has a 2-in-1 PTFE fitting for the injection and extraction of the air sample.
- LDPE bags: these 1.1 L bags are made of low density polyethylene used for filling packaging spaces. The sampling bags are fabricated with a simple valve made from polyethylene as well.

Both, MPE and PVF bags are designed to be filled to 90 % of its volume capacity and every bag has a 2-in-1 PTFE fitting for the injection and extraction of the air sample. Whilst the LDPE bags are fabricated with a valve made from polyethylene as well. The collection of the samples took 3 hours. Each hour we filled 6 sample bags per bag type and carried out the cryogenic sampling with two cold traps. The cryogenic samples were collected through cold traps built with two different pumping rates: 50 mL min$^{-1}$ (slow pumping rate) and 3 L min$^{-1}$ (fast pumping rate). The cold traps were built with a test tube of 50 mL capacity immerse in a container filled with ethanol (100 %) inside a cooler filled with dry ice (-70 ºC). The water collected in both test tubes was thaw and transferred to a 1.5 mL vial for its measurement after the experiment with the LWIA. The cryogenic sampling was performed during the 3 hours of sampling, collecting 3 samples with the FPR and only one with the SPR. We also took our benchmark sample by using the open inlet of the WVIA, collecting 8 samples per hour during the 3 hours experiment (24 samples). During the analysis of the sample bags on days $T_1$, $T_2$, $T_9$, $T_{16}$ and $T_{17}$, we also sampled the laboratory air with one open inlet of the MIU. The samples were analysed and calibrated according to the procedure as described in Section 2.2.1 and Section 2.2.2, using an standard water with an isotope signature of $\delta^{18}$O: -14.44 ‰ and $\delta^2$H: -104.09 ‰.

The consistency analysis of the isotopic signatures was performed comparing the isotope signatures obtained from the different samples against the benchmark collected the same day of the sampling ($T_0$), which corresponds to the direct measurements from the WVIA. The cross comparison was performed with the *Z* analysis (Equation 1) (Orlowski *et al.* 2016, Wassenaar *et al.* 2012). Where *S* is the isotope signature ($\delta^2$H or $\delta^{18}$O) of the bags or cryogenic samples, *B* is the benchmark isotope signature (WVIA) and $\mu$ is the target variability. Differing from Orlowski *et al.* (2016) and Wassenaar *et al.* (2012), the target variability ($\mu$) was stablished as the isotope range measured with the WVIA during the three hours of measurements of the benchmark ($\delta^2$H: 2.0 ‰ and $\delta^{18}$O: 0.4 ‰) considering the transient condition in the laboratory. Thus, we adopted the limits proposed by Orlowski *et al.* (2016) for accurate analysis (*Z-score* < 2.0), questionable analysis (Z-score: 2.0 – 5.0) and unacceptable analysis (Z-score: > 5.0).

$$Z = \frac{S - B}{\mu} \qquad\qquad\qquad \textit{Equation 1}$$

[Figure]

Figure 1. Air sampling bags used to test the consistency among air sampling methods. Silver bag on the left is the MPE bag, the middle transparent bag corresponds to the PVF bag and the transparent bag on the right is the LDPE bags.

**3. Results and Discussion**

Benchmark isotope signature from the 24 measurements performed during the three hours experiment had an isotope signature of -15.61 ± 0.14 ‰ and -115.12 ± 0.47 ‰ for $\delta^{18}O$ and $\delta^2H$, respectively. All the vapor samples collected with the bags that were measured on the same sampling day ($T_0$) are located within the accurate region based on the *Z-score* analysis (Figure 2). Contrary to these samples, the liquid samples collected with the cold traps are located on the unacceptable region, showing *Z* values bigger than 5. The SPR is only based on one sample because we were only able to collect 0.1 mL of liquid during the three hours, while the FPR collected 0.25 mL every hour. Both liquid samples show a heavier signature than the benchmark, as a consequence of incomplete condensation. These differences are linked to not perfect collection efficiencies during the cryogenic sampling with the cold traps (Griffis, 2013).

The signature of the laboratory air changed during the course of our measurements. The orange samples marked as "Laboratory" in Figure 2 depict the differences among the days of measurement and the sampling date ($T_0$). These differences in laboratory air signatures partly influences the measurement results from all the air samples collected with the three types of bags. The MPE samples are the only ones from the experiment with almost all measurements located within the accurate region of the *Z-score* plot (*Z-score* < 2). Despite the accuracy reached with the MPE, the measurements are influenced by the isotope signature of the air within the laboratory. All the measurements after the sampling date with the LDPE and PVF bags are located within the questionable region of the Z-score plot (*Z-score:* 2-5), while the PVF samples from $T_9$ are on the unacceptable region (*Z-score* > 5). These sampling bags are influenced by the isotopic signature of the laboratory air considering its location close to the laboratory signature during the measurements.

Herbstritt *et al*. (2014) depicted the effect of water mass loss by diffusion through the wall of different sampling materials for the determination of stable isotope signatures of soil pore water under equilibrium conditions. They mention the capacity of the layered foil bags (similar material to the MPE bags used in this experiment) to prevent the vapor mass loss during long periods of time. However, the sampling bags design

influences strongly the capacity to retain stable samples. Tock (1983) evaluates the water vapor transmission rate (WVTR) of different polymers films, finding that methalized polyethylene coated materials have the lowest WVTR (1.55 g $m^{-2}d^{-1}$) compared with the LDPE (23.25 g $m^{-2}d^{-1}$) and the PVF (50.22 g $m^{-2}d^{-1}$). The WVTR defines the capacity of a film to transfer water molecules depending on the relative humidity gradient (Kumaran, 1998). Thus, the WVTR of each material provides insights about the variation of the stable isotope measurements, including the MPE bags. It is important to remark that the diffusion characteristics of foil layered materials are directly influence by the temperature in outdoor conditions (Pons *et al.*, 2014).

[Figure]

Figure 2. Dual plot for the *Z-score* values of $\delta^2$H and $\delta^{18}$O of the samples under analysis.

**3. Conclusions**

Water vapor sampling techniques differ in their capacity to keep reliable measurements after the sampling. The MPE bags shows the more accurate measurement of stable isotopes two weeks after its sampling. LDPE and PVF sampling bags can be used for sampling water vapor if the measurements are performed on the same day of sampling. After 24 hours, the WVTR of both materials (LDPE and PVF) allows the air sample to be in equilibrium with the surrounding air affecting the measurement accuracy. Both cold trap systems differ strongly from the benchmark in this experiment. Apparently, even the 3 hours sampling was not sufficient for a full condensation. This long sampling time, limits its use for studying evaporation process that change in short periods of time (< 30min). Thus, underline the need to determine with further research if cryogenic samples

can be used as references or benchmarks when those are compared against direct measurements performed with laser spectrometers as the WVIA.

**References**

Griffis, T. J. (2013) Tracing the flow of carbon dioxide and water vapor between the biosphere and atmosphere: A review of optical isotope techniques and their application, Agricultural and Forest Meteorology, 174-175, 85 – 109. doi: 10.1016/j.agrformet.2013.02.009

Herbstritt, B., Limprecht, M., Gralher, B., Weiler, M. (2014) Effects of soil properties on the apparent water-vapor isotope equilibrium fractionation: Implications for the headspace equilibrium method [poster presentation], UNI Freiburg. Available at: http://www.hydro.uni-freiburg.de/publ/pubpics/post229.

Kumaran, M. (1998) Interlaboratory Comparison of the ASTM Standard Test Methods for Water Vapor Transmission of Materials (E 96-95). Journal of Testing and Evaluation, 26: 83-88. https://doi.org/10.1520/JTE11977J.

Orlowski, N., Pratt, D. L., and McDonnell, J. J. (2016) Intercomparison of soil pore water extraction methods for stable isotope analysis. Hydrol. Process., 30: 3434–3449. doi: 10.1002/hyp.10870

Pons, E., Yrieix, B., Heymans, L., Dubelley, F., & Planes, E. (2014). Permeation of water vapor through high performance laminates for VIPs and physical characterization of sorption and diffusion phenomena. Energy and Buildings, 85, 604-616. Doi: 10.1016/j.enbuild.2014.08.032

Tock, R. W. (1983). Permeabilities and water vapor transmission rates for commercial polymer films. Advances in Polymer Technology: Journal of the Polymer Processing Institute, 3(3), 223-231. doi: 10.1002/adv.1983.060030304

Wassenaar, L. I., Ahmad, M., Aggarwal, P., Duren, M., Pöltenstein, L., Araguas, L. and Kurttas, T. (2012), Worldwide proficiency test for routine analysis of $\delta 2H$ and $\delta 18O$ in water by isotope-ratio mass spectrometry and laser absorption spectroscopy. Rapid Commun. Mass Spectrom., 26: 1641-1648. doi:10.1002/rcm.6270

---

## Author Comment (AC3) · 15 Jan 2019

Thanks for the comments and constructive critics on the section "Specific comments and Technical corrections". We propose some major changes to improve the manuscript according to your recommendations. Additionally, we would like to clarify some misinterpretations or misunderstandings related to specific sections in the paper:

1) However, there appear to be severe misinterpretations of the presented data. Unfortunately, the authors did not compare their sample bag results with data from alternatives of water vapor stable isotope measurements they would have considered trustworthy.

Reply: Regarding the LDPE suitability as sample container, we perform an additional experiment. The full experiment is available as an appendix to this reply and as a proposal to add its results to this manuscript.

In this extra experiment, we compared the consistency of air stored in our LDPE-bag to two commercial air-sampling bags: Tedlar and Foil Bags. We collected air samples and stored them in these different types of bags on 'day 0' and analyzed the samples 1,2,9, 16 and 17 days after collection. The bag samples were also compared to direct measurements of the laboratory air by the WVIA. Additionally, we also weighted the bags to check the conservation of mass.

As the main result, we confirm the low reliability of the LDPE bags when compared to the foil bags and direct measurements with the WVIA. However, the performance of the LDPE is better than the performance of the cold traps. We also found that not only our LDPE bag was affected by the ambient laboratory air, but also the other 2 bags likely due to the water vapor transmission rate of the different materials (Tock, 1983). This confirms the pattern described by Herbstritt et al. (2014) who depicted the effect of water mass loss by diffusion through the wall of different sampling materials for the determination of stable isotope signatures of soil pore water under equilibrium conditions. This experiment and its results are added as supplemental material to the reply to the reviewers.

2) The differences between direct laboratory air measurements and bag-sampled air from the same location (Sample D) are remarkable (Table 1) but ignored in the manuscript.

Reply: Thank you for pointing out this issue. In fact, this interpretation is a misunderstanding from our methods where it was not clearly explained that the sampling date in the laboratory is not the same as when the samples were analyzed. Hence 'sample D' and 'laboratory' should not have a similar isotopic value per se. In method section 2.2.4, page 7, line 3 we omitted to mention that Sample D was collected one week
before the measurement at the laboratory. We added the "laboratory" sample as a background validation during analysis to show the "Laboratory" as a proof of that no laboratory air was leaking and/or mixing within the inlet used to convey the air from the sampling bag to the Multiport Inlet Unit. We agreed that we haven't mentioned the time difference between the sample collection and sample analysis.

Consequently, with the finding from the additional experiment, we decided to remove this section from the experiment to be replaced with the additional experiment (Supplemental Material).

3) The differences between data from bag-sampled air and cold traps are attributed to the alleged failure of the latter. But then why are the authors showing these data? Comparison of vapor concentrations during sampling and during measurements would have been helpful but are missing.

Reply: The additional experiment described in the first section shows the differences between direct measurements performed with the WVIA, 3 different types of bags and cryogenic samples (with 2 different pumping rates) collected with the same cold traps design as the experiment. The cryogenic samples provide a completely different isotope signature as a consequence of the incomplete condensation, even with the slow pumping rate. We believe, it is useful to provide all the collected data even if these ones do not behave as expected. Often cryogenic extraction is used as a benchmark, however, we show that even the slow pumping rate is not enough to reach full condensation. The data about vapor concentrations were not added to keep a reduce the number of figures and focus mainly about their performance during the sampling under field conditions. Additionally, considering the results from the additional experiment this section will be removed from the manuscript.

4) Specifically, I would have expected the cold trap data to follow a trend line, similar to an evaporation line, in dual isotope space as a result of the alleged incomplete vapor sampling. This was not the case (Fig. 8). Neither did they plot towards the upper right

of the sample bag data as must be the case after enrichment in 18O and 2H taking the allegedly unflawed sample bag data as the origin of this evolution. In my perception, the cold trap data may indeed represent the natural variability of sampled air masses. Sample bag isotope data were quite consistent and, moreover, strongly deviating from the cold trap cluster in dual isotope space. However, even if cold trap data were flawed there is no proof that bag samples were not subject to exchange with each other and or via the ambient atmosphere. Conversely, unintended exchange would well explain the similarity of their vapor isotopic compositions. The statement that the mere difference between isotope signatures of laboratory air vapor and bag-sampled vapor is a "good indication" that no exchange occurred is not justified. And it is proven wrong when some of the bag samples were supposed to represent the very laboratory air.

Polyethylene bags similar to the ones used in this study have been shown to allow for evaporative loss of water resulting in measurable changes of the contained water vapor stable isotopic composition within several days of storage (Hendry et al., 2015, doi: 10.5194/hess-19-4427-2015). This happened despite the enclosed water vapor being in isothermal equilibrium with a markedly bigger liquid water reservoir present in the co-enclosed natural soil sample. The vapor-only reservoirs investigated in this study were several orders of magnitude smaller than a typical soil sample liquid water reservoir (microliters vs. milliliters) and must, therefore, be expected to reveal measurable changes in their isotopic compositions within mere minutes. This is the reason why commercially available gas sampling bags, e.g. Lindebags, include one layer of diffusion-tight metal foil.

Reply: Thank you for pointing this out. Based on this comment and your first comment we carried out an extra experiment. Indeed we found that the LDPE bags do not perform well in comparison to our benchmark and the foil bags (MPE). However, the LDPE bags do perform better than the cold traps. The data on this new experiment can be found as a supplement to this reply.

5) In summary, the authors did not demonstrate that sample bag data do in fact represent what they are claimed to represent. Furthermore, I do not see how a re-interpretation of the presented data would suffice the aim of a reliable method for collecting representative discrete water vapor samples. The first two steps of the described experiment are mainly a repetition of the work of Aemisegger et al. (2012, DOI: 10.5194/amt-5-1491-2012) with insufficient novelty to justify their publication.

Reply: We agree that we did not demonstrate that the LDPE bag is a proper sample bag. Hence we altered our scope after we did the additional experiment. Our new scope is to compare different types of collection bags. Although we use partly the methodology of Aemisegger, our study differs from them. The aim of Aemisegger et al. was to demonstrate the application of different analyzers for water vapor sampling, while we focus on the collection bags when direct measurements are not possible (see page 3 line 6-8). We mention the origin of the methodology with the specific references (page 5, line 21), and we use this procedure to determine the shorter time to retrieve a stable measurement from our device.

Reply to Specific Comments: Title: alternative to what? Reply: We propose to change the title to "technical note: comparison of water vapor sampling techniques for stable isotope analysis"

P1-L4: insert "isotopic" before "fractionation" Reply: Done, Thanks.

P1-L5: the quality of the measurement should be characterized because the analyzer will provide continuous data regardless of source. However, only after sufficiently long analysis of a sufficiently large reservoir these data will be e.g. representative, stable, reliable, meaningful, or reasonable. + delete "one" + capacity of what?

Reply: We changed the sentence into "The first experiment determined the minimum air sample volume required to obtain stable measurements of $\delta 2H$ and $\delta 18O$ with a laser spectrometer. The second experiment determined the ability to . . .."

P1-L7: I know "under : : : conditions" but not "under: : :set up". Please rephrase.

Reply: In the final manuscript this sentence is removed

P1-L8: tense: can -> could, allows -> allowed Reply: Done, Thanks.

P1-L11: "resolution", not "variation" Reply: Done, Thanks.

P1-L11: insert "with" before "the cold traps" Reply: Done, Thanks.

P1-L13: given the following sentences, this must be evapotranspiration, not evaporation. What are the provided references referring to? Reply: In this case, we are considering the definition of evaporation provided by Savenije (2004). Thus evaporation is defined as the sum of transpiration, soil and interception evaporation. The provided references refer to the magnitude of terrestrial evaporation on a global scale.

Savenije, H. H.: The importance of interception and why we should delete the term evapotranspiration from our vocabulary. Hydrol. Process., 18: 1507-1511. doi:10.1002/hyp.5563, 2004.

P1-L16: rephrase to e.g. ": : :surfaces. Their partitioning is: : :" or "... surfaces with their partitioning being ..." Reply: We changed this sentence into: " …. by plant and litter surface. The partitioning of evaporation is a key element to understand. . . "

P1-L22: incorrect isotope terminology: delta values do not refer to isotopes but to isotope ratios. Please rephrase. Reply: Rephrased to: "The stable isotope ratios of $\delta2H$:::"

P2-L1: please be more specific: It's the isotope fractionation factors that depend on temperature. Reply: We added: "::: and signature variation due to fractionation is linked to :::"

P2-L1f: rephrase to e.g.: "Physical isotope fractionation is driven by water phase change and also to a lower extent by diffusion." Mixing is a conservative process and does not cause fractionation although it does, in fact, produce a different isotopic composition in the case of two distinct reservoirs being mixed. Reply: Thanks, we

rephrased as follows: "Physical isotope fractionation is driven by water phase changes and also to a lower extent by diffusion while mixing processes can alter the isotopic composition."

P2-L4: delete "whilst" or connect the two sentences Reply: Whilst is replaced by While

P2-L5: "caused by", not "caused during" Reply: Done, Thanks.

P2-L7: "unidirectional"? E.g. net evaporation or net condensation is the result of a mismatch between the absolute evaporation flux and the absolute condensation flux. This makes it highly bidirectional. Reply: We meant that during evaporation the lighter isotopes move from the source towards the atmosphere, which is a unidirectional flux. We understand the confusion, so we decided to remove this second part of the sentence from the manuscript.

P2-L20: start new sentence: "However,: : :" Reply: Done, Thanks.

P2-L30: please rephrase: the risk is not ISOTOPIC fractionation. This is in fact taken into account. The risk is incomplete sampling. Reply: Thank you for this suggestion. We have changed the sentence accordingly.

P3-L1: please quantify "short" Reply: We added the approximate time period.

P3-L2: improvements regarding what? Reply: We referred to improvements regarding the accuracy of the machine.

P3-L6: "inTO the field" Reply: Done, Thanks.

P3-L7: what are "controlled run temperatures"? + insert "apply" after "restrictions" Reply: It should be: "controlled room temperatures". Done, Thanks.

P3-L8: insert "isotope" before "fractionation" Reply: Done, Thanks.

P3-L11: insert "isotope" before "fractionation" Reply: Done, Thanks.

P3-L12: signature -> signatures Reply: This sentence was removed from the
manuscript.

P3-L13: insert "the" before "cold" Reply: This sentence was removed from the manuscript.

P3-L16: please be more specific, e.g. "A LGR (ABB - Los Gatos Research Inc., San Jose, CA, USA) Water Vapor: : :" Reply: We added this extra information to the manuscript.

P3-L16: signature -> signatures Reply: Done, Thanks.

P3-L18: colloquial language, please rephrase, e.g. "with measurements of liquid water standards of known isotopic composition: : :" Reply: Done, Thanks.

P3-L22f: I do not understand this sentence, please rephrase Reply: Rephrase as "In all the measurements, the first MIU inlet was attached to a dried air source. We used this dried air source, which had a distinctly different isotopic signature than the samples, to identify between the different samples of other MIU inlets in the post-processing of the data."

P3-L24: 5000ppm? Dried air should have no more than a few hundred ppm remaining vapor mixing ratio. Please comment on the high number you encountered Reply: This "dried air source" is different from the one used by the WVISS (DAS). Our dried air (not dry) source corresponds to a flux of air not fully dried to provide a different isotope signature during the measurements.

P3-L24f: I do not understand this sentence, please rephrase Reply: We changed the sentence into: "This dried air source was achieved by conveying laboratory air through a 2 L borosilicate bottle that was filled with 1.5 kg of silica gel to dry the laboratory air to a concentration lower than 5000 ppm"

P3-L30: please specify what makes these standard deviations meaningful. For example, are they sufficient to discriminate samples that represent the natural variation of isotope ratios on typical timescales? Reply: These standard deviation values are recommended by Kurita et al (2012) as good measurements during the evaluation of laser spectrometers.

P4-L1: signature -> signatures Reply: Done. Thanks.

P4-L3: analyses, not analysis + what kind of analyses? + tests, not test Reply: Thanks for the recommendation. We refer to all the data analysis. We add "data" in the text.

P4-L6: isotope signatures are expressed in delta values, not just in heavy isotopes + "are" or "were" before "expressed" Reply: The $\delta$ notation is missing here.

P4-L7f: please describe the calibration procedure and the correction – if necessary – of drift and vapor concentration effects during liquid water analyses Reply: We added the following sentence: "The correction and calibration of the isotope signatures of liquid samples were performed with the software LIMS 10.083 (2015)"

P4-L8: please define the abbreviation "IWA". Is this the "WVIA"? Reply: Corrected, it should be WVIA. Thanks.

P4-L15 (and throughout the manuscript): this is a correction, not a calibration, usually resulting in a normalization of raw isotope data to a reasonable water vapor mixing ratio (see e.g. Schmidt et al., 2010, DOI: 10.1002/rcm.4813 or Johnson et al., 2011, DOI: 10.1002/rcm.4894 for more details). Please state why you chose to do differently Reply: We decided to follow the same procedure as Steen-Larsen et al. (2013,2014) and Rambo et al. (2011) because they use similar equipment as we do.

P4-L19: please provide more details on "automatically" + "a" means only one. How many different waters were used for this step? Reply: The WVISS has a software package that provides an automatic calibration according to the settings established by the user. This device allows the use of only one water standard. This water is mentioned on Page 4-line20. We added this information.

P4-L21: I do not understand "water molecule concentrations depending on the air sample concentrations" Reply: This has to do with the specific settings of the device.

The device uses a pump with a specific voltage that allows different water concentration to be injected into the WVIA. So, with the known water signature the device will provide a constant air flow to allow the change of water concentrations (ppm) depending on the isotope water signature. We added this information in the manuscript.

P4-L22: is -> was Reply: Done. Thanks.

P4-L24: are -> were Reply: Done. Thanks.

P4-L24f: add here that these were calculated using equations 2 & 3 + I suggest to not use the symbol alpha as it represents fractionation rather than correction factors in isotope contexts Reply: Thanks for the recommendation. We changed alpha into "$\varphi$".

P4-L27: value -> values + is -> were Reply: Done. Thanks.

P5-L8: was, not were Reply: Done. Thanks.

P5-L12: "run" -> "was run" or better "was conducted" Reply: Done. Thanks.

P5-L16f: this statement should be placed after the description how one minute was determined as ideal aggregation time period. Reply: Done. Thanks.

P5-L18: insert "statistical" or equivalent before "analysis" Reply: Done. Thanks.

P5-L19: isn't it the standard deviation of the moving average that is governed (not driven – sloppy jargon)? + why not connect the two sentences with "and" as they both start with "This analysis"? Reply: We changed the sentence into; "This statistical analysis determines the time interval from where the moving average value is only coming from the white noise of the laser and not by the memory effect of the previous sample. Additionally, the analysis evaluates. . ."

P5-L23: insert "it" before "is" Reply: Done. Thanks.

P5-L25: the Allan deviation plots I know have minima at the respective aggregation time. + Are you referring to Figure 5 here? If so, please state. Furthermore, this figure

and its discussion should appear first, as your first decision (i.e. the 1-min aggregation time) is based on it. Reply: Yes, the Allan deviation does reach the minimum value after the aggregation time of 600 seconds. And we are referring to figure 5, however, the figure location will be solved with the final typesetting.

P6-Figure 1: I do not understand why this effort was necessary. Why wasn't it sufficient to (perform and) look at the 600 s interval to retrieve the desired information? And isn't this information already provided in Aemisegger et al. (2012, DOI: 10.5194/amt-5-1491-2012) or could have been concluded from the injection frequency and valve operation pattern of routine liquid water analyses performed on such analyzers? Similar objections apply for the aggregation time.

Reply: Aemisegger et al. (2012) determine the minimum time required to have reliable measurements from an outdoor system (direct measurements from the atmosphere). In our case, we follow the same procedure to determine the minimum time required for an indoor system with less variations. In our case, the variability depends on the memory effect and dead volume along the tubing. We perform the experiment in this way to determine if smaller sampling times will trigger changes in isotope signatures due to the no reduction of the memory effect along the tubing. Longer volumes of air guarantee the reduction, but small volumes won't allow the reduction of the memory effect.

P6-L5: why were polyethylene bags selected despite being aware of the findings of Hendry et al. (2015, doi: 10.5194/hess-19-4427-2015)? See general comments for details. Reply: Response to this can be found in the previous section on General Comments.

P6-L6ff: such paragraphs should be written in past tense Reply: Done. Thanks.

P6-L11: it has -> with Reply: Done. Thanks.

P6-L12: insert "for" before "the tight" + what do you mean by "movement"? Reply:

Done. I do mean connector.

P7-L1: "Air samples were collected manually..." Reply: Done. Thanks.

P7-L6: why didn't you normalize all measurements to e.g. 10k ppm i.e. calculate the raw isotope numbers the analyzer would have shown if the vapor concentration had been 10k ppm (see e.g. Schmidt et al., 2010, DOI: 10.1002/rcm.4813 or Johnson et al., 2011, DOI: 10.1002/rcm.4894 for details), prior to calibration?

Reply: The device has an internal calibration for 10K ppm. However, it is necessary to perform the calibration proposed by Steen-Larsen et al. (2013,2014) and Rambo et al. (2011) due to the internal drift of the WVIA.

P7-L8: statistical -> statistically significant Reply: Done. Thanks.

P7-L11: commonly, the deuterium excess is indicated by the lower case letter d (in italics) Reply: Done. Thanks.

P8-L7: an -> a Reply: Done. Thanks.

P8-L11: were -> was Reply: Done. Thanks.

P8-L13: it would be important to read that the tubes reached the bottom of the bottles in a way that only the minimized inner cross section area of the tubes allowed for the interfacial exchange between sampled water and ambient atmosphere. Were they installed that way? 15 cm sounds a little short for 5-L bottles. And 9 mm sounds a little wide for this purpose. How would this affect your LMWL? Reply: This is a typo on the manuscript. The bottle is 2.5 L. Thanks for the observation.

P8-L14: 6mm inner or outer diameter? Both of which appear quite a lot. + "reduce the vapour exchange" -> "facilitate pressure compensation while at the same time minimizing loss via vapor diffusion" or equivalent. Pressure compensation is necessary once the inlet tube is submersed into the sampled water which should happen as soon as possible (see previous comment)

Reply: It was 6 mm of outer diameter and we follow the description giving by Gröning et al (2012).

P8-L18: simultaneously to the cold trap or to each other? Reply: It should be "simultaneously to the cold trap". Thanks for the suggestion.

P8-L19: why not 24? (4 h * 6 samples/h = 24 samples) Reply: Yes, it is 24 samples.

P8-L21: please rephrase and start a new sentence (the frozen vapor was not closed nor did it collect...), e.g. "The liquid water sample was immediately transferred into..." Reply: Done. Thanks.

P8-L23: delete "a" Reply: Done. Thanks.

P8-L25: "the" not "its", because vapor was measured, not vapor condensation or sampling bags Reply: Done. Thanks.

P8-L25f: I do not understand this statement. Wasn't the concentration at which the samples were analyzed just the one present inside the bags? Reply: Yes, but it is necessary to perform the calibration because of the WVIA drift as it was mentioned previously on the section 2.2.2. We changed 'analysis' for 'calibration'.

P9-Figure 3: this figure should appear in the method section. Throughout the manuscript, all figures should appear near their description. Reply: This issue depends on the Latex processor and can be solved during typesetting.

P9-L8: insert "probably" before "because", as this is your speculation Reply: Done. Thanks.

P9-L11: please start new sentence ("However,: : :") + I do not understand "some averages with non stable measurements" Reply: We refer to those measurements with a standard deviation bigger than the defined thresholds. We clarified this in the manuscript.

P9-L12: please make sure that figures and their description and discussion appear in

the right order Reply: This issue depends on the Latex processor and can be solved during typesetting.

P10-Figure 4: see comment on P6-Figure 1 Reply: Done. Thanks.

P10-Figure 5: I am unable to find 0.3‰ or 1.5‰ on the vertical axis, thus I am unable to see what aggregation time is sufficient to reach these standard deviations + the numbers on the vertical axis are not evenly spaced + the label of the horizontal axis should be "aggregation time" or equivalent + this figure and its discussion should appear before any figure featuring 1-min-means because those were chosen based on this analysis of the Allan deviation + Reply: In the Allan Deviation graph, we have to see the stable line on the plot and not the Standard Deviation, because the procedure to estimate those ones is completely different. "moving", not "mobile". Reply: Thanks, Done.

P10-L1: aren't 450 mL calculated quite tightly? What if you have two strongly differing successive samples and the memory effect causes the readings from the second sample to not have stabilized after 240s leaving not enough time for a 1-min-average before the bag is empty? Reply: Here, we are specifying the minimum volume required to provide a reliable measurement with the standard deviation thresholds established for both isotopes. Additionally, the reason for providing a "dried air" (not dry air) was to provide a different isotope signature between measurements to determine the memory effect of a different sample.

Further, the smaller the vapor reservoir, the higher its susceptibility to contamination Reply: The definition of a minimum air volume required provides information about the type of sampling bag is needed. So, the user can determine later on according with their project restrictions the volume to be sampled in the field.

+ delete "to carry out 300 s of continuous measurements" as you provided this number in the previous sentence already. Reply: Thanks for the recommendation.

P11-L2: I do not understand why this step was necessary. Do you have indication that small vapor reservoirs such as your sampling bags would reveal significant variations? If so please elaborate on this also in the introduction Reply: We just want to confirm the aggregation time

P11-L4: insert "probably" before "because", as this is your speculation Reply: Done. Thanks.

P11-L6: whose capacity? Reply: This should be "ability".

P11-L8: signature -> signatures + insert "those of" after "from" Reply: Done. Thanks.

P11-L8f: I strongly disagree with this statement. From your experimental design and data, there is no way of telling whether your bag samples are or are not a mixture of the original (e.g. flux tower) sample and other sources. This would only have been possible if you had analyzed a distinct air directly and sampled it into bags in parallel, then stored the bags while exposing them to a different ambient atmosphere, then analyzed the bag air, and then compared the results of direct and discrete sample measurements. Reply: This issue about the LDPE bags suitability for sampling in the field is discussed on the supplemental experiment. We have to conclude that the LDPE is indeed not suitable for collecting water vapor for longer than 1 day. Therefore we removed experiment 3 from the manuscript.

P11-L10: statistical -> statistically significant + insert "and" between the two delta expressions Reply: Done. Thanks.

P11-L11: insert "probably" before "the reason", as this is your speculation Reply: Done. Thanks.

P11-L12: insert "probably" before "because", as this is your speculation + are you referring to the absolute deviation of your arithmetic mean from the true value (i.e. the accuracy) or are you rather referring to the standard deviation (i.e. the precision)? + on -> of Reply: Thanks for the recommendation.

P11-L13: showed -> observed Reply: Done. Thanks.

P11-L14: is this the within-sample or the between-sample deviation? Reply: This is between samples. Thanks.

P11-L15: why would wind change the isotopic composition? Are you referring to different air parcels being sampled? Reply: Considering the forest matrix where the tower is placed, low wind conditions will allow a stronger signature from the Douglas Fir transpiration. Whilst with stronger wind, the footprint will retrieve isotope signatures from other blocks mixed with the signature from the forest stand under analysis. However, we decided to delete this experiment from the manuscript.

P11-L16: sampling -> sample + per set of sample or per sample? Reply: Done. Thanks.

P12-Table 1: shouldn't laboratory air and sample D be consistent? Could it be that the sample bags were stored in a confined space where they exchanged with each other? The consistency among A-D is striking. And so is the discrepancy between laboratory air and sample D. Why was sample D not discussed in the manuscript? This could have been an indication that bag samples represent what they are supposed to represent. In order for a potential consistency of lab air and D to be a proof, conditions as described in comment to.

Reply: We agree that this was unclear. Please have a look at our reply in the general comments where we clarify what we did.

P11-L8f would have been necessary + in the figure caption: lower case -> superscript + this needs more details. What exactly is different when a, b, c, or d is displayed? The note at the bottom of the table states that lower case letters on the same column are statistically different. Each letter represents a homogeneous group determined with the Tukey test.

P12-L4: delete "or after" or write "or later on that day" Reply: Done. Thanks.

P12-L5f: given, that you report the LGR measurements in ppm, can you provide ppm values for the observed humidity as well? This might give the reader a clue whether your sampling was conservative or exchange with ambient air has occurred.

Reply: Thanks for the recommendation. Unfortunately, we don't have data on the humidity in the laboratory. However, we changed experiment 2 completely by comparing different sample bags in time to a benchmark and the open inlet.

P12-L10: the offset of the equation has the "unit" ‰ Reply: Corrected. Thanks.

P12-L11: is located -> plots + "heavier" is too colloquial, please rephrase Reply: Done. Thanks.

P12-L12: insert "isotopic signatures" before "water vapor Reply: Done. Thanks.

P12-L13: "lighter" is too colloquial, please rephrase + insert delta symbols before 2H and 18O Reply: Done. Thanks.

P12-L14: insert "show" before "less variation" Reply: Done. Thanks.

P12-L14f: but shouldn't it be the opposite? Cold trap samples should represent a mixture of six potentially variable bag samples. The similarity among air samples leads me to the conclusion that the originally present natural variation, still revealed to some degree by the cold trap data, got completely lost when all bag samples exchanged with or via a similar atmosphere prior to analysis Reply: This is explained in the additional experiment.

P12-L15f: the location of atmospheric vapor isotope signatures relative to the LMWL also depend on the slope thereof. Reply: We decided to remove experiment 3

P12-L17f: aren't these interpretations referring to liquid water? You are showing vapor data. Therefore, you first have to determine where the corresponding liquid water reservoir would plot relative to the LMWL before making these statements Reply: We refer to the cold traps. Considering the issues of incomplete condensation showed with

the additional experiment. We decided to remove experiment 3

P12-L21: insert "isotope" before "signature" Reply: We decided to remove experiment 3

P12-L22: heavier delta2H -> higher delta2h values + incomplete at -70_C? What was the remaining vapor pressure at the cold trap outlet? Reply: We decided to remove experiment 3

Assuming that cold trap data might be flawed, why did you present them as a reference for the sample bag data? Reply: We showed this as a comparative experiment. However, we did not expect the issues with the cold traps and neither the LDPE bags. We decided to remove experiment 3

Why was it not possible to design the experiment in a way that the reference data set (i.e. cold trap) is trustworthy? Further, wouldn't incomplete condensation result in a trend line extending to the upper right of the sample bag data rather than in a data cloud located towards the upper left? + enrichment of what?

Reply: The enrichment in $\delta$2H shows the data above the sampling bags. This pattern was seen as well with the additional experiment added as supplemental material to this response. Therefore we decided to remove experiment 3.

P12-L23: delete "it" + replace "is" by "may be", as this is your speculation and strongly depends on setup properties Reply: We decided to remove experiment 3

P13-Figure 7: I suggest "time of day (hh:mm)" as label of the horizontal axis Reply: We decided to remove experiment 3

P13-L3: all -> the entire + exception from big differences? + is shown -> was observed Reply: We decided to remove experiment 3

P13-L4: "Isotope signature at 34m height from cold traps: : :" -> "Isotope signatures of samples collected at 34 m height via cold traps: : :" Reply: We decided to remove

experiment 3

P13-L4f: what if these data were the true numbers and your bag sample data were flawed? Reply: After the additional experiment, we check the incomplete condensation of the cryogenic samples and the issues from the LDPE bags. We decided to remove experiment 3

P13-L7: enriched in what? Reply: We decided to remove experiment 3

P13-L12: what was the air flow rate through the cold traps? This information would be useful to calculate the minimum humidity (in ppm) during sampling. Reply: The air flow through the cold traps is specified on section 2.2.5 (3 L min-1). We decided to remove experiment 3

Based on this estimate, further interpretations of the sample bags' diffusion-tightness might be possible. Reply: Thanks for the recommendation. We did this in our new experiment 2, which is added as supplementary text to this reply. Based on these results we decided to change our scope and remove experiment 3.

P14-L7: insert "it" after "experiment 2" and "isotope" after "stable" Reply: Done. Thanks.

P14-L7f: please rephrase this sentence Reply: Done. Thanks.

P14-L10: resolution, not variation + please connect the two sentences or rephrase Reply: Done. Thanks.

P14-L12: please connect the two sentences or rephrase Reply: Done. Thanks.

P14-L14: dew temperature -> dew point Reply: Done. Thanks.

P14-L15: of -> or + insert "the dew point in" before "the field". Reply: Done. Thanks. Do you have a suggestion how to deal with this situation? Reply: Not yet.

P15-Figure 9: proportional to -> indicating Reply: The figure caption states: "proportional to the standard deviation of each sample". However, we decided to remove experiment 3.

Please also note the supplement to this comment:
https://www.hydrol-earth-syst-sci-discuss.net/hess-2018-538/hess-2018-538-AC3-supplement.pdf